# A TRUST REGION APPROACH FOR FEW-SHOT SIM-TO-REAL REINFORCEMENT LEARNING

## ABSTRACT

Simulation-to-Reality Reinforcement Learning (Sim-to-Real RL) seeks to use simulations to minimize the need for extensive real-world interactions. Specifically, in the few-shot off-dynamics setting, the goal is to acquire a simulator-based policy despite a dynamics mismatch that can be effectively transferred to the real-world using only a handful of real-world transitions. In this context, conventional RL agents tend to exploit simulation inaccuracies resulting in policies that excel in the simulator but underperform in the real environment. To address this challenge, we introduce a novel approach that incorporates a penalty to constrain the trajectories induced by the simulator-trained policy inspired by recent advances in Imitation Learning and Trust Region based RL algorithms. We evaluate our method across various environments representing diverse Sim-to-Real conditions, where access to the real environment is extremely limited. These experiments include high-dimensional systems relevant to real-world applications. Across most tested scenarios, our proposed method demonstrates performance improvements compared to existing baselines.

## 1 INTRODUCTION

Reinforcement Learning (RL) is often applied in simulation before deploying the learned policy on real systems (Ju et al., 2022; Muratore et al., 2019; Kaspar et al., 2020; Witman et al., 2019). This approach is considered to be one of the safest and most efficient ways of obtaining a near-optimal policy for complex systems (Jiang et al., 2021; Salvato et al., 2021; Hsu et al., 2023), as many of the challenges of applying RL to real-world systems (Dulac-Arnold et al., 2021) are mitigated. The agent can sample the simulator at will (Kamthe & Deisenroth, 2018; Schwarzer et al., 2021) without having to consider any safety constraints (García & Fernández, 2015; Achiam et al., 2017) during training.

However, simulators of complex systems are often inaccurate. Indeed, many physical laws such as contact forces, material elasticity, and fluid dynamics are difficult to model, leading simulators to rely on approximations (Koenig & Howard, 2004; Todorov et al., 2012). These small inaccuracies may accumulate over time, leading to increasing deviations between the simulated and real-world environments over time. Directly transferring a policy trained on simulators can therefore lead to unsatisfactory outcomes due to these compounding errors. Worse, modern optimization-based agents may exploit these discrepancies to find policies that perform exceptionally well in simulation but result in trajectories that are impossible to replicate in the real environment. This phenomenon - known as the *Simulation-to-Reality* (Sim-to-Real) gap (Höfer et al., 2021) - occurs in most simulators (Salvato et al., 2021).

In general, besides relying only on a simulator, it is still possible to deploy the agent in the environment to collect data. However, this deployment is limited due to safety and time considerations. As a result, the available data is often limited to a few narrow trajectories. Two orthogonal approaches are possible to include this data in the derivation of the policy. The first one - well studied (Abbeel et al., 2006; Zhu et al., 2018; Desai et al., 2020a; Hanna et al., 2021) - leverages this data to improve the simulator, then learns a traditional RL agent on the upgraded simulator.

The second approach keeps the simulator fixed and biases the learning process to account for the dynamics discrepancies (Koos et al., 2012). This line of work is complementary to the one improving the simulator, as both could be combined to make the best use of the limited real-world samples. To the best of our knowledge, only a few works have taken this purely off-dynamics direction, and even

fewer have focused on the low data regime scenario. Currently, the prominent approach is DARC (Eysenbach et al., 2020) which modifies the reward function to search for parts of the simulator that behave similarly to the real world. Although this method is effective for a few classes of problems, e.g. with the "broken" environments, we have found that it may fail drastically in others, limiting its application to a restrictive class of discrepancy between the simulator and the environment.

In this paper, we introduce the **F**ew-sh**O**t **O**ff **D**ynamics (FOOD) algorithm, a Trust Region method constraining the derived policy to be around the trajectories observed in the real environment. We theoretically justify this constraint, which directs the policy towards feasible trajectories in the real system, and thus mitigates the potential trajectory shifts towards untrustable regions of the simulator. Our constraint takes the form of a regularization between visitation distributions, we show that it can be practically implemented using state-of-the-art techniques from the Imitation Learning (IL) literature (Hussein et al., 2017). Our method is validated on a set of environments with multiple off-dynamics disparities. We show that, compared to other baselines, our approach is the most successful at taking advantage of the few available data. Our agent is also shown to be relevant for a wider range of dynamics discrepancies.

## 2 RELATED WORK

Closing the Sim-to-Real gap for the dynamics shift is crucial for the successful deployment of RL-based policies in real-world systems. In this scenario, the algorithms trained on the simulator must have sufficient performance in the real world to have a tangible impact. This problem has been studied in two contexts, depending on the accessibility of the agent to transitions from the real environment. These settings are referred to as "Zero-Shot" and "Few-Shot" Sim-to-Real.

**Zero-Shot Sim-to-Real RL** Sampling data from real-world environments can be impossible due to strict safety constraints or time-consuming interactions. In such cases, simulators are used to ensure robustness (Morimoto & Doya, 2005; Moos et al., 2022) to guarantee a certain level of performance without sampling from the real system. It can take many forms. One possible choice is domain randomization (Mordatch et al., 2015; Tobin et al., 2017; Peng et al., 2018; Mehta et al., 2020) where relevant parts of the simulator are randomized to make it resilient to changes. Another line of work focuses on addressing the worst-case scenarios under stochastic simulator dynamics (Abdullah et al., 2019; Tanabe et al., 2022). Robustness can also be achieved w.r.t. actions (Jakobi et al., 1995; Pinto et al., 2017; Tessler et al., 2019), that arise when certain controllers become unavailable in the real environment. These techniques are outside the scope of this paper as they do not involve any external data in the learning process.

**Few-Shot Sim-to-Real RL** When data can be sampled from the real environment, two orthogonal approaches have been developed to bridge the Sim-to-Real gap. The first approach, well established, is to improve the accuracy of the simulator through various methods. If the simulator parameters are available, the simulator parameters can be optimized directly (Farchy et al., 2013; Zhu et al., 2018; Tan et al., 2018; Collins et al., 2021; Allevato et al., 2020; Du et al., 2021). Otherwise, expressive models can be used to learn the changes in dynamics (Abbeel et al., 2006; Saveriano et al., 2017; Lee et al., 2017; Golemo et al., 2018; Hwangbo et al., 2019). Within this category, a family of methods builds an action transformation mechanism that - when taken in simulation - produces the same transition that would have occurred in the real system (Hanna & Stone, 2017; Karnan et al., 2020; Desai et al., 2020b; Hanna et al., 2021). In particular, GARAT (Desai et al., 2020a) leverages recent advances in Imitation Learning from Observations (Torabi et al., 2018; 2019) to learn this action transformation and ground the simulator with only a few trajectories. All these algorithms are orthogonal to our work as once the simulator has been improved, a new RL agent has to be trained.

The second approach, more related to our work, is the line of inquiry that alters the learning process of the RL policy in the simulator to be efficient in the real environment. One group of approaches creates a policy - or policies - that can quickly adapt to a variety of dynamic conditions (Yu et al., 2018; Arndt et al., 2020; Yu et al., 2020; Kumar et al., 2021). It requires the ability to set the simulator parameters which may not always be feasible, e.g. if the simulator is a black box. A more general algorithm is DARC (Eysenbach et al., 2020). It learns two classifiers to distinguish transitions between the simulated and real environments and incorporates them into the reward function to account for the dynamics shift. Learning the classifiers is easier than correcting the dynamics of the simulator, but

as we will see in the experiments, this technique seems to work mainly when some regions of the simulator accurately model the target environment and others don't. Another related work is H2O (Niu et al., 2022a) which extends the approach by considering access to a fixed dataset of transitions from a real environment. It combines the regularization of the offline algorithm CQL (Kumar et al., 2020; Yu et al., 2021; Daoudi et al., 2022) with the classifiers proposed by DARC. However, the performance of H2O is dependent on the amount of data available. In fact, it performed similarly, or worse, to the pure offline algorithm when only a small amount of data was available (Niu et al., 2022a, Appendix C.3).

## 3 BACKGROUND

### 3.1 PRELIMINARIES

Let $\Delta(\cdot)$ be the set of all probability measures on $(\cdot)$. The agent-environment interaction is modeled as a Markov Decision Process (MDP) $(\mathcal{S}, \mathcal{A}, r, P, \gamma, \rho_0)$, with a state space $\mathcal{S}$, an action space $\mathcal{A}$, a transition kernel $P : \mathcal{S} \times \mathcal{A} \to \Delta(\mathcal{S})$, a reward function $r : \mathcal{S} \times \mathcal{A} \times \mathcal{S} \to [R_{\min}, R_{\max}]$, the initial state distribution $\rho_0$ and a discount factor $\gamma \in [0, 1)$. A policy $\pi : \mathcal{S} \to \Delta(\mathcal{A})$ is a decision rule mapping a state over a distribution of actions. The value of a policy $\pi$ is measured through the value function $V_P^\pi(s) = \mathbb{E}_{\pi,P}\left[\sum_{t=0}^\infty \gamma^t r(s_t, a_t, s_{t+1})|s_0 = s\right]$. The objective is to find the optimal policy maximizing the expected cumulative rewards $J_P^\pi = \mathbb{E}_{\rho_0}\left[V_P^\pi(s)\right]$. We also define the $Q$-value function $Q_P^\pi(s, a) = \mathbb{E}_{\pi,P}\left[\sum_{t=0}^\infty \gamma^t r(s_t, a_t, s_{t+1})|s_0 = s, a_0 = a\right]$ and the advantage value function $A_P^\pi(s, a) = Q_P^\pi(s, a) - V_P^\pi(s)$. Finally, let $d_P^\pi(s) = (1 - \gamma)\mathbb{E}_{\rho_0,\pi,P}\left[\sum_{t=0}^\infty \gamma^t \mathbb{P}(s_t = s)\right]$, $\mu_P^\pi(s, a) = (1 - \gamma)\mathbb{E}_{\rho_0,\pi,P}\left[\sum_{t=0}^\infty \gamma^t \mathbb{P}(s_t = s, a_t = a)\right]$ and $\nu_P^\pi(s, a, s') = (1 - \gamma)\mathbb{E}_{\rho_0,\pi,P}\left[\sum_{t=0}^\infty \gamma^t \mathbb{P}(s_t = s, a_t = a, s_{t+1} = s')\right]$ the state, state-action and state-action-state visitation distributions. All these quantities are expectations w.r.t. both the policy and the transition probabilities.

The Sim-to-Real problem involves two MDPs: the source simulator $\mathcal{M}_s$ and the target environment $\mathcal{M}_t$. We hypothesize that the simulator and the real world are identical except for their transition probabilities $P_s \neq P_t$. Our hypothesis states that while most of the MDP parameters are known, the underlying physics of the environment is only estimated. This is a common setting known as off-dynamics Reinforcement Learning. It encapsulates many real-world applications: a model of the dynamics may have been previously learned, or practitioners may have created a simulator based on a simplification of the system's physics. We do not assume access to any parameter modifying the transition probabilities $P_s$ to encompass black-box simulators. For readability purposes, we drop the $P$ subscripts for the value functions as they are always associated with the source simulator $\mathcal{M}_s$.

Many Few-Shot Sim-to-Real agents (Abbeel et al., 2006; Desai et al., 2020a; Hanna et al., 2021) typically employ the following procedure to handle complex environments. First, the policy and value functions are initialized in the simulator. The choice of objective at this stage can vary, although a classical approach is to solely maximize the rewards of the simulator. At each iteration, the policy is verified by experts. If it is deemed safe, $N$ trajectories are gathered in the real environment and saved in a replay buffer $\mathcal{D}_t$. These trajectories are then used to potentially correct the simulator and/or the training objective and induce a new policy. This process is repeated until a satisfactory policy is found. This setup is time-consuming and may be risky even when the policy is verified by experts, hence the need to learn with as few data as possible from the real environment.

This work focuses on how to best modify the objective with few trajectories. If handled properly, this could reduce the number of interactions required by the whole process overcoming the need to build a perfect simulator. For the purpose of our study, we assume that $\mathcal{M}_t$ remains fixed throughout the process.

### 3.2 TRUST REGION ALGORITHMS AND THE VISITATION DISTRIBUTION CONSTRAINT

Due to their efficiency and stability, Trust Region algorithms (Schulman et al., 2015; 2017; Kumar et al., 2020) have shown strong efficiency in various RL settings. Many of them are based on Kakade & Langford (2002), where an iteration scheme improves the policy by maximizing an estimate of the Advantage function within a Trust Region (Yuan, 2000) around the previous policy. This process has been extended by refining the Trust Region (Terpin et al., 2022; Moskovitz et al., 2020), for example

by introducing a behavior $b_P^\pi$ (Mouret, 2011) that encapsulates any additional property of the MDP (Pacchiano et al., 2020; Touati et al., 2020).

This family of algorithms is formalized as follows. A policy and a value function are parametrized with respective weights $\theta \in \Theta$ and $\omega \in \Omega$, that we denote from now on $\pi_\theta$ and $V_\omega^{\pi_\theta}$. At each iteration $k$, the policy is improved using the advantage function built from the approximated value function $V_{\omega_k}^{\pi_{\theta_k}}$ while ensuring that the policy remains in the Trust Region:

$$
\begin{aligned}
&\underset{\theta \,\in\, \Theta}{\text{maximize}} && \mathbb{E}_{s \sim d_P^{\pi_{\theta_k}}(\cdot), a \sim \pi_\theta(\cdot|s)} \left[ A_{\omega_k}^{\pi_{\theta_k}}(s, a) \right] \\
&\text{subject to} && D\left( b_P^{\pi_\theta} \,\Big\|\, b_P^{\pi_{\theta_k}} \right) \leq \epsilon_k,
\end{aligned}
\tag{1}
$$

where $D$ is any kind of similarity metric and $\epsilon_k$ is a hyper-parameter. TRPO (Schulman et al., 2015; Shani et al., 2020) can be retrieved with $b_P^\pi = \pi$ and by setting $D$ to be the Kullback-Leibler (KL) divergence. Alternative behavior options can be found in (Pacchiano et al., 2020; Touati et al., 2020; Moskovitz et al., 2020). In particular, Touati et al. (2020) proposed to encapsulate the whole trajectories induced by $\pi$ and $P$ by setting $b_P^\pi = d_P^\pi$. It resulted in better results both in terms of sample efficiency and final cumulative rewards than most of its counterparts. This is natural as the new constraint between the state visitation distributions takes the whole trajectories induced by the policy into account, providing more information than the policy alone.

## 4 FEW-SHOT OFF DYNAMICS REINFORCEMENT LEARNING

In this section, we propose a new objective to better transfer a policy learned in simulation to the real environment. We extend the Trust Region Policy Optimization objective to the off-dynamics setting. Then, we remind necessary results on Imitation Learning (IL) before deriving our practical algorithm **F**ew-sh**O**t **O**ff **D**ynamics (FOOD) Reinforcement Learning.

### 4.1 A NEW SIM-TO-REAL TRUST REGION OBJECTIVE

Given the discrepancies between the dynamics of the simulator and the real environment, applying the same policy to both environments may result in different trajectories. This poses a challenge as the agent may make the most of these differences to find policies that produce excellent trajectories in the simulator but are impossible to replicate in the real system.

We first analyze the difference between the objectives $J_P^\pi$ associated with the target and source environments, depending on a metric between visitation distributions. For this, we apply the tools from traditional Trust Region methods (Pirotta et al., 2013; Schulman et al., 2015; Achiam et al., 2017) to the Sim-to-Real setting, and propose the following lower bound.

**Proposition 4.1** *Let $J_P^\pi = \mathbb{E}_{\rho_0}\left[V_P^\pi(s)\right]$ the expected cumulative rewards associated with policy $\pi$, transitions $P$ and initial state distribution $\rho_0$. For any policy $\pi$ and any transition probabilities $P_t$ and $P_s$, the following holds:*

$$
J_{P_t}^\pi \geq J_{P_s}^\pi - \frac{2R_{max}}{1 - \gamma} D_{TV}\left(\nu_{P_s}^\pi, \nu_{P_t}^\pi\right),
\tag{2}
$$

*with $D_{TV}$ the Total Variation distance $D_{TV}\left(\nu_{P_s}^\pi, \nu_{P_t}^\pi\right) = \sup_{s,a,s'} \left| \nu_{P_s}^\pi(s, a, s') - \nu_{P_t}^\pi(s, a, s') \right|$.*

We defer the proof to Appendix A. It illustrates how the performance of the optimal policy in the real environment may differ from that of the simulator due to the metric $D_{\mathrm{TV}}\left(\nu_{P_s}^\pi, \nu_{P_t}^\pi\right)$ quantifying their difference in trajectories. This is all the more important given that this term is exacerbated by the factor $\frac{2R_{\max}}{1-\gamma}$. Finally, we note that the Total Variation distance could be replaced by the Kullback-Leibler divergence or by the Jensen-Shannon divergence using Pinsker's inequality (Csiszar & Körner, 1981) or the one in (Corander et al., 2021, Proposition 3.2), provided the minimal assumptions of having a finite state-action space and the absolute continuity of the considered measures.

Overall, this lower bound highlights a good transfer between the source and target environment is possible when $D_{\text{TV}}\left(\nu_{P_s}^{\pi}, \nu_{P_t}^{\pi}\right)$ is small, as it induces similar objectives $J_P^{\pi}$. Inspired by this insight, we adapt Trust Region methods to the Sim-to-Real setting (Pirotta et al., 2013; Schulman et al., 2015; Achiam et al., 2017) and propose a new constraint between trajectories by setting the behaviors $b_P^{\pi}$ to be the state-action-state visitation distribution respectively associated with the transition probabilities of the source and target environment $\nu_P^{\pi}$:

$$
\begin{aligned}
&\underset{\theta \in \Theta}{\text{maximize}} \quad \mathbb{E}_{s \sim d_{P_s}^{\pi_{\theta_k}}(\cdot), a \sim \pi_\theta(\cdot|s)}\left[A_{\omega_k}^{\pi_{\theta_k}}(s, a)\right] \\
&\text{subject to} \quad D\left(\nu_{P_s}^{\pi_\theta} \,\Big\|\, \nu_{P_t}^{\pi_{\theta_k}}\right) \leq \epsilon_k.
\end{aligned}
\tag{3}
$$

The new constraint ensures that the policy is optimized for trajectories that are feasible in the real world, thus preventing the RL agent from exploiting any potential hacks that may exist in the simulator. In addition, remaining close to the data sampled from the real world can be beneficial when the simulator has been constructed using that data, as querying out-of-distribution data can yield poor results (Kang et al., 2022).

Unfortunately, the difference between the transition probabilities makes the constraint in Equation 3 difficult to compute. The previous work of Touati et al. (2020) addressed this by restricting $D$ to $f$-divergences $D_f\left(\mu_P^{\pi_\theta} \| \mu_P^{\pi_{\theta_k}}\right) = \mathbb{E}_{(s,a) \sim \pi_\theta}\left[f\left(\frac{\mu_P^{\pi_\theta}}{\mu_P^{\pi_{\theta_k}}}\right)\right]$ and by considering state-action visitation distributions. They Touati et al. (2020) used the DualDICE algorithm (Nachum et al., 2019) to directly estimate the relaxed ratio $\frac{\mu_P^{\pi_\theta}}{\mu_P^{\pi_{\theta_k}}}$ for any policy $\pi_\theta$ sufficiently close to $\pi_{\theta_k}$, eliminating the need to sample data for each policy. However, this method is not applicable to our setting because DualDICE relies on a modified joined Bellman operator, which assumes that both distributions follow the same transition probabilities. Another solution would be to collect at least one trajectory per update. While this would not pose any safety concerns for the data would be sampled in the simulator, it can be time-consuming in practice.

## 4.2 PRACTICAL ALGORITHM

In order to devise a practical algorithm for addressing Equation 3, we solve the penalized problem

$$
\underset{\theta \in \Theta}{\text{maximize}} \quad \mathbb{E}_{s \sim d_{P_s}^{\pi_{\theta_k}}(\cdot), a \sim \pi_\theta(\cdot|s)}\left[A_{\omega_k}^{\pi_{\theta_k}}(s, a) + \alpha\, \mathcal{R}(s, a)\right],
\tag{4}
$$

where the regularization $\mathcal{R}(s, a)$ serves as a proxy for minimizing the divergence $D\left(\nu_{P_s}^{\pi_\theta} \,\Big\|\, \nu_{P_t}^{\pi_{\theta_k}}\right)$ and $\alpha$ is a hyper-parameter.

To construct a relevant proxy, we leverage the recent results from Imitation Learning (IL) (Hussein et al., 2017) that we briefly recall in this paragraph. In this field, the agent aims to reproduce an expert policy $\pi_e$ using limited data sampled by that expert in the same MDP with generic transition probabilities $P$. Most current algorithms tackle this problem by minimizing a certain similarity metric $D$ between the learning policy's state-action visitation distribution $\mu_P^{\pi_\theta}$ and the expert's $\mu_P^{\pi_e}$. The divergence minimization problem is transformed into a reward $r_{\text{imit}}$ maximization one, resulting in an imitation value function $V_{\text{imit}}^{\pi} = \mathbb{E}_{\pi, P_s}\left[\sum_{t=0}^{\infty} \gamma^t r_{\text{imit}}(s_t, a_t, s_{t+1})|s_0 = s\right]$. Since these algorithms are based on data, they can be used to minimize the chosen similarity metric $D$ between two state-action-state visitation distributions with different transition probabilities. Applied to our setting, this is formalized as:

$$
\arg\max_{\pi} V_{\text{imit}}^{\pi} = \arg\min_{\pi} D\left(\nu_{P_s}^{\pi_\theta} \,\Big\|\, \nu_{P_t}^{\pi_{\theta_k}}\right).
\tag{5}
$$

The choices for the divergence $D$ are numerous, leading to different IL algorithms (Ho & Ermon, 2016; Fu et al., 2017; Xiao et al., 2019; Dadashi et al., 2020), some of which are summarized in Table 1.

| GAIL | AIRL | PWIL |
|---|---|---|
| $D_{\text{JS}}\left(X_{P_{\text{s}}}^{\pi_\theta} \,\middle\|\, X_{P_{\text{t}}}^{\pi_{\theta_k}}\right)$ | $D_{\text{KL}}\left(X_{P_{\text{s}}}^{\pi_\theta} \,\middle\|\, X_{P_{\text{t}}}^{\pi_{\theta_k}}\right)$ | $D_{\text{W}}\left(X_{P_{\text{s}}}^{\pi_\theta}, X_{P_{\text{t}}}^{\pi_{\theta_k}}\right)$ |

Table 1: Objective function for well-known Imitation Learning (IL) algorithms. We included their general form with $X$, that can be chosen as either $d$, $\mu$, or $\nu$. Other IL agents can be found in (Ghasemipour et al., 2020).

These IL techniques enable efficient estimation of this value function using a small number of samples from $d_{P_{\text{t}}}^{\pi_{\theta_k}}$ and unlimited access to $\mathcal{M}_{\text{s}}$. Let $\xi \in \Xi$ be the weights of this parametrized value function. The new regularization is $\mathcal{R}(s,a) = A_{\text{imit}}^{\pi_{\theta_k},\xi_k}(s,a)$, which can be learned with any suitable IL algorithm.

This new agent is quite generic as it could be optimized with different divergences. It takes as input an online RL algorithm (Babaeizadeh et al., 2016; Schulman et al., 2017) denoted $\mathcal{O}$ and an Imitation Learning algorithm denoted $\mathcal{I}$. The whole Sim-to-Real algorithm process, which we denote Few-shOt Off Dynamics (FOOD) RL, is described as follows. First, the policy and the value weights are initialized in the simulator with $\mathcal{O}$. At each iteration $k$, the agent samples $N$ new trajectories with $\pi_{\theta_k}$[1]. Subsequently, the policy, traditional, and imitation value functions are retrained on the simulator with $\mathcal{O}$ and $\mathcal{I}$ according to Equation 4. The whole algorithm is summarized in Algorithm 1.

---

**Algorithm 1** Few-shOt Off Dynamics (FOOD)

---

**Input:** Algorithms $\mathcal{O}$ and $\mathcal{I}$
Initialize policy and value weights $\theta_0$ and $\omega_0$ with $\mathcal{O}$
Randomly initialize the weights $\xi_0$
**for** $k \in (0,\dots,K-1)$ **do**
    Gather $N$ trajectories $\{\tau_i,\dots,\tau_N\}$ with $\pi_{\theta_k}$ on the real environment $\mathcal{M}_t$ and add them in $\mathcal{D}_{\text{t}}$
    Remove trajectories that lead to drastic failures
    Learn the value function weights $\omega_{k+1}$ with $\mathcal{O}$ in the source environment $\mathcal{M}_{\text{s}}$
    Learn the imitation value function weights $\xi_{k+1}$ with $\mathcal{I}$ in $\mathcal{M}_{\text{s}}$ using $\mathcal{D}_{\text{t}}$
    Learn the policy maximizing equation 4 using $\mathcal{D}_{\text{t}}$ and $\mathcal{M}_{\text{s}}$ with $\mathcal{O}$
**end for**

---

## 5 EXPERIMENTS

In this section, we evaluate the performance of the FOOD algorithm in the off-dynamics setting in environments presenting different dynamics discrepancies, treated as black box simulators. These environments are based on Open AI Gym (Brockman et al., 2016) and the Minitaur environment (Coumans & Bai, 2016–2021) where the target environment has been modified by various mechanisms. These include gravity, friction, and mass modifications, as well as broken joint(s) systems for which DARC is known to perform well (Eysenbach et al., 2020, Section 6). We also add the Low Fidelity Minitaur environment, highlighted in previous works (Desai et al., 2020a; Yu et al., 2018) as a classical benchmark for evaluating agents in the Sim-to-Real setting. In this benchmark, the source environment has a linear torque-current relation for the actuator model, and the target environment - proposed by Tan et al. (2018) - uses accurate non-linearities to model this relation.

All of our FOOD experiments were carried out using both GAIL (Ho & Ermon, 2016), a state-of-the-art IL algorithm, as $\mathcal{I}$. We found that GAIL performed similarly, or better than other IL algorithms such as AIRL (Fu et al., 2017) or PWIL (Dadashi et al., 2020). FOOD is tested with its theoretically motivated metric between state-action-state visitation distributions $\nu_P^\pi$, as well as with $d_P^\pi$ and $\mu_P^\pi$ for empirically analyzing the performance associated with the different visitation distributions. We found that GAIL performed similarly, or better than other IL algorithms such as AIRL (Fu et al., 2017) or PWIL (Dadashi et al., 2020). The performance of our agent with the different IL algorithms can be found in Appendix C.5. We compare our approach against various baselines modifying the RL objective, detailed below. They cover current domain adaptation, robustness, or offline Reinforcement

---
[1]These trajectories could first be used to improve the simulator.

Learning techniques applicable to our setting. Further details of the experimental protocol can be found in Appendix C.

- **DARC** (Eysenbach et al., 2020) is our main baseline. It is a state-of-the-art off-dynamics algorithm that introduces an additional importance sampling term in the reward function to cope with the dynamics shift. In practice, this term is computed using two classifiers that distinguish transitions from the simulated and the real environment. In this agent, an important hyper-parameter is the standard deviation $\sigma_{\text{DARC}}$ of the centered Gaussian noise injected into the training data to stabilize the classifiers (Eysenbach et al., 2020, Figure 7). DARC was originally optimized with SAC (Haarnoja et al., 2018) but to allow a fair comparison with FOOD, we re-implemented DARC in the same RL algorithm that is used in our method, drawing inspiration from the open-source code (Niu et al., 2022b).

- **Action Noise Envelope (ANE)** (Jakobi et al., 1995) is a robust algorithm that adds a centered Gaussian noise with standard deviation $\sigma_{\text{ANE}}$ to the agent's actions during training. Although simple, this method outperformed other robustness approaches in recent benchmarks (Desai et al., 2020a) when the simulator is a black box.

- **CQL** (Kumar et al., 2020) is an offline RL algorithm that learns a policy using real-world data. It does not leverage the simulator in its learning process, which should make its resulting policy sub-optimal. This algorithm inserts a regularization into the $Q$-value functions, with a strength $\beta$. We use Geng (2021) to run the experiments.

- We also consider two RL agents, **RL$_{\text{Sim}}$** trained solely on the simulator (without access to any real-world data) and **RL$_{\text{Real}}$** trained solely on the real-world environment. Both algorithms were trained to convergence. Even though the latter baselines do not fit in the off-dynamics setting they give a rough idea of how online RL algorithms would perform in the real environment. The online RL algorithm $\mathcal{O}$ depends on the environment: we use A2C (Babaeizadeh et al., 2016) for Gravity Pendulum and PPO (Schulman et al., 2017) for the other environments.

**Experimental protocol**   Our proposed model and corresponding off-dynamics/offline baselines require a batch of real-world data. To provide such a batch of data, we first train a policy and a value function of the considered RL agent until convergence by maximizing the reward on the simulated environment. After this initialization phase, 5 trajectories are sampled from the real environment to fit the restricted real data regime. They correspond to 500 data points for Pendulum and 1000 data points for the other environments. If some trajectories perform poorly in the real environment, we remove them for FOOD, DARC, and CQL to avoid having a misguided regularization. FOOD, DARC, and ANE are trained for 5000 epochs in the simulated environment. Both RL$_{\text{Sim}}$ and RL$_{\text{Real}}$ are trained until convergence. CQL is trained for 100000 gradient updates for Gravity Pendulum and 500000 gradient updates for all other environments. All algorithms are averaged over 4 different random seeds. Additional details can be found in Appendix C.

**Hyperparameters Optimization**   We optimize the hyperparameters of the evaluated algorithms through a grid search for each different environment. Concerning DARC and ANE, we perform a grid search over their main hyper-parameter $\sigma_{\text{DARC}} \in \{0.0, 0.1, 0.5, 1\}$ and $\sigma_{\text{ANE}} \in \{0.1, 0.2, 0.3, 0.5\}$. The remaining hyperparameters were set to their default values according to the experiments reported in the open-source code (Niu et al., 2022b). For CQL, we perform a grid search over the regularization strength $\beta \in \{5, 10\}$, otherwise we keep the original hyper-parameters of Geng (2021). For RL$_{\text{Real}}$ and RL$_{\text{Sim}}$ we used the default parameters specific to each environment according to Kostrikov (2018) and trained them over 4 different seeds. We then selected the seed with the best performance. For our proposed algorithm FOOD, the regularization strength hyperparameter $\alpha$ is selected over a grid search depending on the underlying RL agent, $\alpha \in \{0, 1, 5, 10\}$ for A2C and $\alpha \in \{0.5, 1, 2, 5\}$ for PPO. This difference in choice is explained by the fact that the advantages are normalized in PPO, giving a more refined control over the regularization weight.

**Results**   We monitor the evolution of the agents' performance by evaluating their average return $\mathcal{R}$ in the real environment during training. Note that we do not gather nor use the data from those evaluations in the learning process since we work under a few-shot framework. We compute the return of all methods averaged over 4 seeds, where we show the standard deviation being divided

by two for readable purposes. In all figures, the $x$-axis represents the number of epochs where each epoch updates the policy and value functions with $8$ different trajectories from the simulator.

## 5.1 COMPARISON BETWEEN THE DIFFERENT AGENTS

We evaluate the mentioned algorithms on the proposed environments. These experiments provide an overview of the efficiency of the different objectives in finetuning the policy, given reasonably good trajectories. Results are summarized in Table 2, where we also report the median of the normalized average return (NAR) $\frac{J_{P_t}^{\pi_{\text{agent}}} - J_{P_t}^{\pi_{\text{RL-Sim}}}}{J_{P_t}^{\pi_{\text{RL-Real}}} - J_{P_t}^{\pi_{\text{RL-Sim}}}}$ (Desai et al., 2020a) as well as the median of the NAR's standard deviations. The associated learning curves and the NAR per environment can be found in Appendix C.2.

| Environment | RL$_{\text{Sim}}$ | RL$_{\text{Real}}$ | CQL | ANE | DARC | FOOD (Ours) | | |
| --- | --- | --- | --- | --- | --- | --- | --- | --- |
| | | | | | | $\mu_P^{\tilde{\pi}}$ | $d_P^{\tilde{\pi}}$ | $\nu_P^{\tilde{\pi}}$ |
| Gravity Pendulum | $-1964 \pm 186$ | $-406 \pm 22$ | $-1683 \pm 142$ | $-2312 \pm 11$ | $-3511 \pm 865$ | $-2224 \pm 43$ | $\mathbf{-485 \pm 54^*}$ | $-2327 \pm 14$ |
| Broken Joint Cheetah | $1793 \pm 1125$ | $5844 \pm 319$ | $143 \pm 104$ | $3341 \pm 132$ | $2553 \pm 405$ | $\mathbf{3801 \pm 155}$ | $3888 \pm 201$ | $\mathbf{3950 \pm 97^*}$ |
| Broken Joint Minitaur | $7.4 \pm 4.1$ | $20.8 \pm 4.8$ | $0.25 \pm 0.09$ | $7.8 \pm 6$ | $\mathbf{12.9 \pm 2}$ | $\mathbf{14.9 \pm 3}$ | $\mathbf{13.6 \pm 3.8}$ | $\mathbf{16.6 \pm 4.7^*}$ |
| Heavy Cheetah | $3797 \pm 703$ | $11233 \pm 1274$ | $41 \pm 34$ | $\mathbf{7443 \pm 330^*}$ | $3956 \pm 1314$ | $4876 \pm 181$ | $4828 \pm 553$ | $4743 \pm 297$ |
| Broken Joint Ant | $5519 \pm 876$ | $6535 \pm 352$ | $1042 \pm 177$ | $3231 \pm 748$ | $5041 \pm 364$ | $\mathbf{6145 \pm 98}$ | $5547 \pm 204$ | $\mathbf{6179 \pm 86^*}$ |
| Friction Cheetah | $1427 \pm 674$ | $9455 \pm 3554$ | $-466.4 \pm 13$ | $\mathbf{6277 \pm 1405^*}$ | $3064 \pm 774$ | $3890 \pm 1495$ | $3212 \pm 2279$ | $3852 \pm 733$ |
| Low Fidelity Minitaur | $8.9 \pm 5.8$ | $27.1 \pm 8$ | $10.2 \pm 1$ | $6.4 \pm 3$ | $3.2 \pm 1.8$ | $\mathbf{17 \pm 2}$ | $\mathbf{15.7 \pm 2.8}$ | $\mathbf{17.9 \pm 0.8^*}$ |
| Broken Leg Ant | $1901 \pm 981$ | $6430 \pm 451$ | $830 \pm 8$ | $\mathbf{2611 \pm 220}$ | $\mathbf{2336 \pm 565}$ | $\mathbf{2652 \pm 356}$ | $2345 \pm 806$ | $\mathbf{2733 \pm 719^*}$ |
| Median NAR and std | $0 \; ; \; 0.25$ | $1 \; ; \; 0.26$ | $-0.32 \; ; \; 0.02$ | $0.1 \; ; \; 0.11$ | $0.06 \; ; \; 0.13$ | $\mathbf{0.37 \; ; \; 0.09}$ | $\mathbf{0.29 \; ; \; 0.17}$ | $\mathbf{0.40 \; ; \; 0.06^*}$ |

Table 2: Returns over 4 seeds of the compared methods on benchmark settings. The best agent w.r.t. the mean is highlighted with boldface and an asterisk. We perform an unpaired t-test with an asymptotic significance of 0.1 w.r.t. the best performer and highlight with boldface the ones for which the difference is not statistically significant.

All the experiments clearly demonstrate the insufficiency of training traditional RL agents solely on the simulator. The optimal policy for the simulator is far from optimal for the real world as we observe a large drop in performance from RL$_{\text{Real}}$ to RL$_{\text{Sim}}$ on all benchmarked environments. For example, the RL$_{\text{Sim}}$ exploits the linear torque-current relation in Low Fidelity Minitaur and fails to learn a near-optimal policy for the real environment. Furthermore, RL$_{\text{Sim}}$ often exhibits a large variance in real environments as it encounters previously unseen situations. This is welcome as relevant trajectories are gathered to guide the agent in the simulation.

Overall, we can see that our algorithm FOOD exhibits the best performances across all considered environments against all other baselines, whether it is constrained by state, state-action or state-action-state visitation distributions. Two exceptions are on Heavy and Friction Cheetah where ANE has very good results. We also note that FOOD with its theoretically motivated regularization between the state-action-state visitation distribution provides the best results with a lower variance.

In addition, we find that the prominent baseline DARC is not efficient in all the use cases. It seems to be particularly good at handling sharp dynamics discrepancies, e.g. when one or two joints are broken but struggles for more subtle differences. In fact, it deteriorates over the naive baseline RL$_{\text{Sim}}$ by a large margin for the Gravity Pendulum and the Low Fidelity Minitaur environments. This may be explained by their reward modification $\Delta_r$ (see Appendix B.1) which prevents the agent from entering dissimilar parts of the simulator but seems unable to handle simulators with a slight global dynamics mismatch. Even when DARC improves over RL$_{\text{Sim}}$, our algorithm FOOD is able to match or exceed its performance. The robust agent ANE is a strong baseline in most environments but may degrade the performance of traditional RL agents, as seen in Low Fidelity Minitaur, Broken Joint Ant, and Gravity Pendulum. CQL did not provide any good results, except on Gravity Pendulum and Low Fidelity Minitaur, but this was to be expected given the few real trajectories the agent has access to. Finally, we note that the three agents FOOD, DARC, and ANE often reduce the variance originally presented in RL$_{\text{Sim}}$.

We attribute FOOD's success to its ability to force the simulator to improve the rewards of the simulator along real-world trajectories. Its regularization seems to be efficient even in the low data regime we are studying.

## 5.2 HYPERPARAMETER SENSITIVITY ANALYSIS

We previously reported the results of the best hyper-parameters of the different methods. In practice, it is important to have a robust range of hyper-parameters in which the considered method performs well. Indeed, to the best of our knowledge, there currently exists no accurate algorithm for selecting such hyper-parameters in a high dimensional environment when the agent has access to limited data gathered with a different policy (Fu et al., 2020). In this section, we detail the sensitivity of FOOD associated with PPO and GAIL to its hyper-parameter $\alpha$ in 3 environments. They were specifically chosen to illustrate the relevant range for $\alpha$. FOOD's complete hyper-parameter sensitivity analysis, as well as the one of DARC and ANE, can respectively be found in Appendix C.4, Appendix C.6 and Appendix C.7.

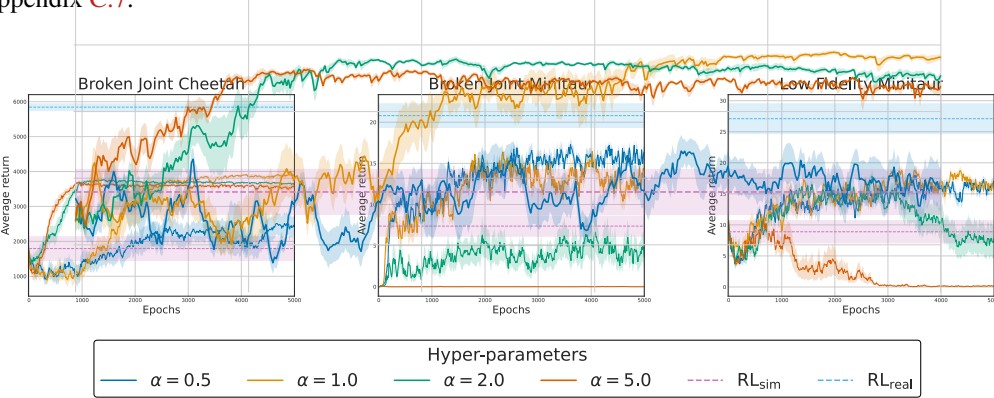

Figure 1: Hyperparameter sensibility analysis for FOOD on three environments.

The hyper-parameter $\alpha$ controls the strength of the regularization in FOOD. If it is too low, the agent will focus mainly on maximizing the rewards of the simulator and becomes very close to the naive baseline $RL_{Sim}$. This can be seen in Broken Joint Cheetah. On the other hand, setting $\alpha$ to a high value may induce the agent to solely replicate the trajectories from the real environment, in which case the agent may also be close to $RL_{Sim}$. Even worse, a hard regularization may degrade over $RL_{Sim}$, as shown in Low Fidelity Minitaur for $\alpha = 5$. However, $5$ is an extremely high value as advantages are normalized in PPO, and this may increase the gradient too much and corrupt learning.

In any case, we have found that FOOD provides the best results when the regularization has the same scale as the traditional objective. This is also verified for the environments not displayed in this sub-section. We conclude that FOOD is relatively robust to this range of hyper-parameter, and recommend using PPO with $\alpha$ close to $1$, a natural choice given that PPO normalizes its advantage functions.

## 6 CONCLUSION

In this work, we investigated different objectives to optimize a policy in different few-shot off-dynamics Sim-to-Real scenarios, including the state-of-the-art method DARC. We found that these objectives are either too simplistic or unable to cope with complex dynamics discrepancies, thereby limiting their application to real-world systems. To address this challenge, we introduced a novel trust region objective along with a practical algorithm leveraging imitation learning techniques. Through experimentations in different Sim-to-Real use cases, we have shown that our approach often outperforms the existing methods and seems to be more robust to dynamics changes. Our results emphasize the importance of leveraging a few real-world trajectories for a proper simulation-to-reality transfer with a well-defined objective.

Our agent could also benefit from new advances in the Imitation Learning literature to gain control in building its Trust Region. Finally, this Trust Region can be useful when the simulator has been improved using the available real-world trajectories as it avoids querying the simulator for Out-of-Distribution samples. This will be the primary focus of our future work.

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
