# A   PROOFS

In this section, we present the proof of our Proposition 4.1 that we restate below, as well as its extensions using different discrepancy measures.

**Proposition A.1** *Let $\nu_P^\pi(s, a, s')$ the state-action-state visitation distribution, where $\nu_P^\pi(s, a, s') = (1 - \gamma)\mathbb{E}_{\rho_0, \pi, P}\left[\sum_{t=0}^{\infty} \gamma^t \mathbb{P}\left(s_t = s, a_t = a, s_{t+1} = s'\right)\right]$. For any policy $\pi$ and any transition probabilities $P_t$ and $P_s$, the following holds:*

$$J_{P_t}^\pi \geq J_{P_s}^\pi - \frac{2R_{max}}{1 - \gamma} D_{TV}\left(\nu_{P_s}^\pi, \nu_{P_t}^\pi\right), \tag{6}$$

*with $D_{TV}$ the Total Variation distance.*

**Proof A.1** *It is known that $J_P^\pi = \frac{1}{1-\gamma}\mathbb{E}_{\nu_P^\pi}\left[r(s, a, s')\right]$. Now:*

$$\left|J_{P_t}^\pi - J_{P_s}^\pi\right| = \frac{1}{1 - \gamma}\left|\left(\mathbb{E}_{\nu_{P_t}^\pi}\left[r(s, a, s')\right] - \mathbb{E}_{\nu_{P_s}^\pi}\left[r(s, a, s')\right]\right)\right| \tag{7}$$

$$= \frac{1}{1 - \gamma}\left|\int_{s,a,s'} \left(r(s, a, s')\nu_{P_t}^\pi(s, a, s') - r(s, a, s')\nu_{P_s}^\pi(s, a, s')\right) \mathrm{d}\left\{sas'\right\}\right| \tag{8}$$

$$= \frac{1}{1 - \gamma}\left|\int_{s,a,s'} r(s, a, s')\left(\nu_{P_t}^\pi(s, a, s') - \nu_{P_s}^\pi(s, a, s')\right) \mathrm{d}\left\{sas'\right\}\right| \tag{9}$$

$$\leq \frac{2R_{max}}{1 - \gamma} D_{TV}\left(\nu_{P_s}^\pi, \nu_{P_t}^\pi\right). \tag{10}$$

*The last inequality is an application of Holder's inequality, by setting $p$ to $\infty$ and $q$ to 1.*

An application of Pinsker inequality (Csiszar & Körner, 1981) provides a similar upper bound with the ullback Leibleir divergence.

**Proposition A.2** *Let $\nu_P^\pi(s, a, s')$ the state-action-state visitation distribution, where $\nu_P^\pi(s, a, s') = (1 - \gamma)\mathbb{E}_{\rho_0, \pi, P}\left[\sum_{t=0}^{\infty} \gamma^t \mathbb{P}\left(s_t = s, a_t = a, s_{t+1} = s'\right)\right]$. For any policy $\pi$ and any transition probabilities $P_t$ and $P_s$ such that $\nu_{P_s}^\pi$ is absolutely continuous with respect to $\nu_{P_t}^\pi$, the following holds:*

$$J_{P_t}^\pi \geq J_{P_s}^\pi - \frac{\sqrt{2}R_{max}}{1 - \gamma}\sqrt{D_{KL}\left(\nu_{P_s}^\pi \,\|\, \nu_{P_t}^\pi\right)}, \tag{11}$$

*with $D_{KL}$ the Kullback Leibleir divergence.*

A lower bound with the Jensen Shannon divergence can also be found thanks to Corander et al. (2021, Proposition 3.2).

**Proposition A.3** *We assume the state-action space. Let $\nu_P^\pi(s, a, s')$ the state-action-state visitation distribution, where $\nu_P^\pi(s, a, s') = (1 - \gamma)\mathbb{E}_{\rho_0, \pi, P}\left[\sum_{t=0}^{\infty} \gamma^t \mathbb{P}\left(s_t = s, a_t = a, s_{t+1} = s'\right)\right]$. We assume the support of $\nu_{P_s}^\pi$ and $\nu_{P_s}^\pi$ is $\mathcal{S} \times \mathcal{A} \times \mathcal{S}$. Then, for any policy $\pi$ and any transition probabilities $P_t$ and $P_s$, the following holds:*

$$J_{P_t}^\pi \geq J_{P_s}^\pi - \frac{4R_{max}}{(1 - \gamma)}\sqrt{D_{JS}\left(\nu_{P_s}^\pi \,\|\, \nu_{P_t}^\pi\right)}, \tag{12}$$

*with $D_{JS}$ the Jensen Shannon divergence.*

# B   ALGORITHMS DETAILS

In this section, we further present the different algorithms used in this paper.

## B.1 DOMAIN ADAPTATION WITH REWARDS FROM CLASSIFIERS (DARC)

We introduce our main baseline Domain Adaptation with Rewards from Classifiers (DARC), which is the prominent state-of-the-art algorithm that tackles the Sim-to-Real task by modifying the RL objective.

DARC takes a variational perspective to the off-dynamics RL problem. Given a trajectory $\tau = (s_0, a_0, s_1, a_1, \dots)$, the target distribution $p(\tau)$ over trajectories is defined as the trajectories maximizing the exponentiated rewards in the real environment:

$$p(\tau) = \rho(s_0) \left( \prod_t P_{\mathrm{t}}(s_{t+1}|s_t, a_t) \right) \exp \left( \sum_t r(s_t, a_t) \right). \tag{13}$$

Let the agent's distributions over trajectories in the simulator $q^{\pi_\theta}(\tau)$ be:

$$q^{\pi_\theta}(\tau) = \rho(s_0) \left( \prod_t P_{\mathrm{s}}(s_{t+1}|s_t, a_t) \right) \pi_\theta(a_t|s_t). \tag{14}$$

DARC minimizes the reversed KL-divergence between $q^{\pi_\theta}(\tau)$ and $p(\tau)$, which results in the following objective expression:

$$-D_{\mathrm{KL}}(q^{\pi_\theta}(\tau) \parallel p(\tau)) = \mathbb{E}_{\tau \sim q^{\pi_\theta}(\cdot)} \left[ \sum_{t=1}^{T} r(s_t, a_t) + \mathcal{H}\left(\pi_\theta(\cdot|s_t)\right) + \Delta r(s_t, a_t, s_{t+1}) \right], \tag{15}$$

with $\Delta r(s_t, a_t, s_{t+1}) = \log P_{\mathrm{t}}(s_{t+1}|s_t, a_t) - \log P_{\mathrm{s}}(s_{t+1}|s_t, a_t)$ and $\mathcal{H}(\cdot)$ the entropy.

The additional reward term incentivizes the agent to select transitions from the simulator that are similar to the real environment. We hypothesize that this is why DARC may struggle when the discrepancies between the simulated and real environment are global. Since the transition probabilities are unknown, DARC uses a pair of binary classifiers to infer whether transitions come from the simulated or real environment. These classifiers are then used to create a proxy equivalent to $\Delta r$.

## B.2 GENERATIVE ADVERSARIAL IMITATION LEARNING (GAIL)

Generative Adversarial Imitation Learning (GAIL) (Ho & Ermon, 2016) is a state-of-the-art Imitation Learning algorithm. Its goal is to recover an expert policy $\pi_e$ by minimizing the Jensen-Shanon divergence between the state-action visitation distributions of the expert and the learning policy. Both the expert and the learning policy interact with the same MDP with transition probabilities $P$. For ease of notation, let $\mathbb{E}_\pi [c(s, a)] = \mathbb{E} \left[ \sum_{t=0}^{\infty} \gamma^t c(s, a)|s_0 \sim \rho, a_t \sim (\cdot|s_t), s_{t+1} \sim P(\cdot|s_t, a_t) \right]$ for any cost function $c : \mathcal{S} \times \mathcal{A} \to \mathbb{R}$.

The authors define the general objective to solve by introducing a convex cost function regularizer $\psi : \mathbb{R}^{S \times A} \to \mathbb{R}$ and its convex conjugate $\psi^*$:

$$\underset{\theta \in \Theta}{\mathrm{minimize}} \quad \psi^*(\mu_P^{\pi_\theta} - \mu_P^{\pi_e}) - \mathcal{H}(\pi_\theta). \tag{16}$$

Following Equation 13 of Ho & Ermon (2016) which defines $\psi_{\mathrm{GAIL}}$, the authors establish the following equivalence:

$$\psi_{\mathrm{GAIL}}^*(\mu_P^{\pi_\theta} - \mu_P^{\pi_e}) = \sup_{D \in (0,1)^{S \times A}} \mathbb{E}_{\pi_\theta} \left[ \log \left( D(s, a) \right) \right] + \mathbb{E}_{\pi_e} \left[ \log \left( 1 - D(s, a) \right) \right] \tag{17}$$

where $D : \mathcal{S} \times \mathcal{A} \to (0, 1)$ is a classifier. Finally, it is demonstrated this specific convex cost function induces the following objective:

$$\underset{\theta \in \Theta}{\text{minimize}} \quad \psi^*_{\text{GAIL}}(\mu^{\pi_\theta}_P - \mu^{\pi_e}_P) - \mathcal{H}(\pi_\theta) = D_{\text{JS}}(\mu^{\pi_\theta}_P \parallel \mu^{\pi_e}_P) - \mathcal{H}(\pi_\theta). \tag{18}$$

In practice, the classifier $D$ is trained to distinguish between samples $(s, a) \in (\mathcal{S} \times \mathcal{A})$ from $\mu^{\pi_\theta}_P$ and $\mu^{\pi_e}_P$. The reward used for optimizing the RL agent is given by $r_{\text{imit}} = -\log(D(s, a))$. If the classifiers learn to distinguish between state samples, then the algorithm minimizes $D_{\text{JS}}(d^{\pi_\theta}_P \parallel d^{\pi_e}_P) - \mathcal{H}(\pi_\theta)$.

### B.3 Conservative Q-Learning (CQL)

In the offline setting, agents aim to learn a good policy from a fixed data set of $M$ transitions $\mathcal{D} = \{(s_i, a_i, s_{i+1})\}^M_{i=0}$ that was collected with an unknown behavioral policy $\pi_\beta$. Offline RL algorithms have demonstrated impressive results when the data set is gathered with a sufficiently good policy and possesses enough transitions, often outperforming the behavioral policy.

Conservative Q-Learning (CQL) (Kumar et al., 2020) is a state-of-the-art offline RL algorithm. It modifies the learning procedure of the $Q$-functions to favor transitions appearing in the data set. At iteration $k$, the $Q$-values are updated as follows:

$$\underset{\omega \in \Omega}{\text{minimize}} \quad \beta \mathbb{E}_{s \sim \mathcal{D}} \left[ \left( \log \sum_{a \in \mathcal{A}} \exp\left( Q^{\pi_{\theta_k}}_\omega(s, a) \right) - \mathbb{E}_{a \sim \pi_\beta(\cdot|s)} \left[ Q^{\pi_{\theta_k}}_\omega(s, a) \right] \right) \right] + \mathcal{E}\left( Q^{\pi_{\theta_k}}_\omega \right), \tag{19}$$

where $\mathcal{E}(Q)$ represents the traditional loss associated with the $Q$-functions. The regularization, controlled by the hyper-parameter $\beta$, penalizes the $Q$-values associated with state-action pairs not appearing in the data set.

## C Experimental Details

In this section, in addition to the values of the hyperparameters necessary to replicate our experiments, we provide further details of the experimental protocol and training. In this section, considering the possible high variance of $\text{RL}_{\text{Sim}}$, the standard deviation is multiplied by a factor of $0.3$. The original variance can be found in Table 2.

### C.1 Environment Details

In all the considered environments, one property is modified in the real environment.

**Gravity Pendulum**   Gravity is increased to $14$ instead of $10$. Since the pendulum requires more time to reach the objective, we also increase the length of each episode to $500$ time-steps in the real environment, while keeping the original length of $200$ time-steps in the simulator.

**Broken Joint or Leg environments**   In these environments, the considered robot - either HalfCheetah or Ant - is crippled in the target domain, where the effect of one or two joints is removed. In practice, this means that it sets one or two dimensions of the action to $0$. These environments were extracted from the open source code of (Eysenbach et al., 2020).

**Heavy Cheetah**   The total mass of the HalfCheetah MuJoCo robot is increased from $14$ to $20$.

**Friction Cheetah**   The friction coefficient of the HalfCheetah MuJoCo robot's feet is increased from $0.4$ to $1$.

**Low Fidelity Minitaur**   The original Minitaur environment uses a linear torque-current linear relation for the actuator model. It has been improved in (Tan et al., 2018) by introducing non-linearities into this relation where they managed to close the Sim-to-Real gap for a real Minitaur environment. In practice, the Minitaur environment can be found in the PyBullet library (Coumans & Bai, 2016–2021). The high fidelity is registered as MinitaurBulletEnv-v0. The low fidelity environment can

be recovered by calling MinitaurBulletEnv-v0 and by setting the argument `accurate motor model enabled` to False and `pd control enabled` to True.

## C.2 LEARNING CURVES AND NORMALIZED RETURN

We report here the learning curves of the different agents mentioned in this paper. For clarity purposes, we keep all baselines fixed except for our agent and DARC, our main competitor. Here, FOOD uses the regularization with $d_P^\pi$ for Gravity Pendulum and $\nu_P^\pi$ for the other environments as GAIL proved to be more stable when FOOD used PPO.

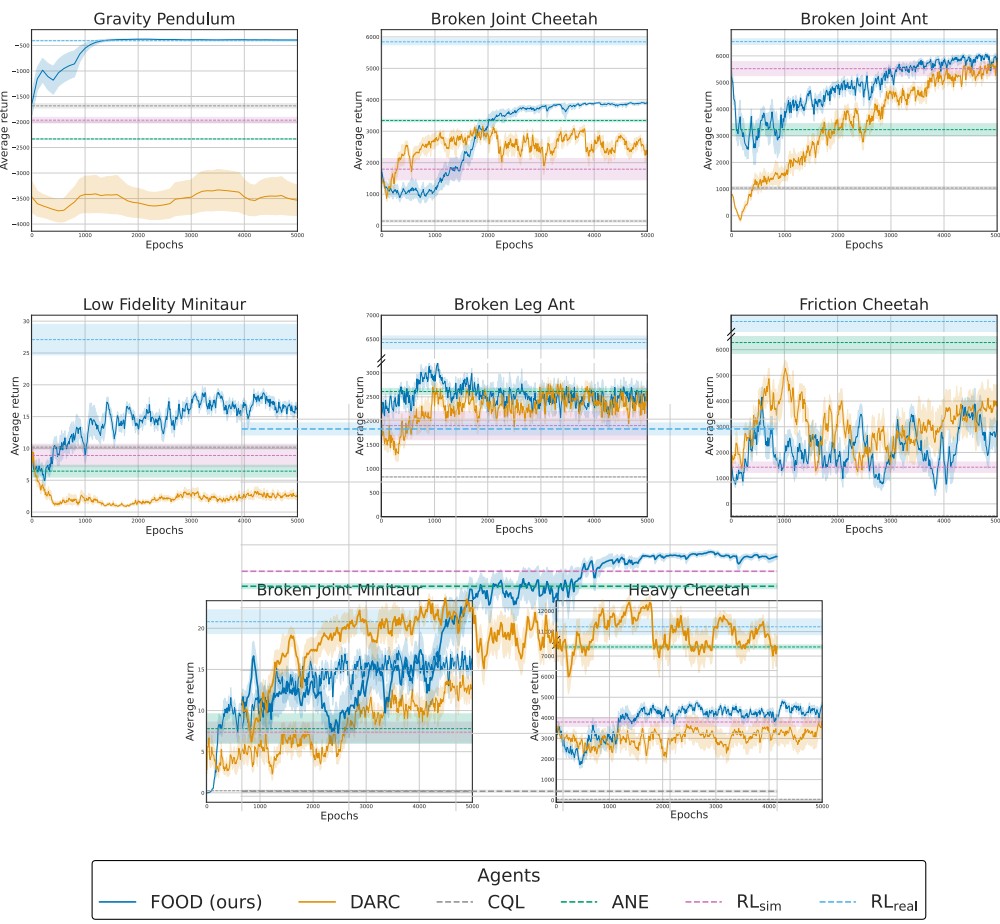

Figure 2: Learning curves of FOOD and DARC for all the proposed environments.

Finally, we report the normalized return of all environments in Table 3.

## C.3 GLOBAL HYPER-PARAMETERS

Our experiments are based on the A2C and PPO implementations proposed by the open-source code (Kostrikov, 2018). We also found that it may be profitable to add a TanH function at the end of the network's policy for the PPO agent. We have selected their hyper-parameters according to the source (Raffin, 2020) and included them in the table below.

**The Minitaur environments** As proposed by the PyBullet library (Coumans & Bai, 2016–2021), $\gamma$ is set to 0.995 for the Minitaur environments. Besides, unlike the Gym and Mujoco environments, they do not use a Tanh squashing function in their policy and the `num-processes` hyper-parameter is set to 1.

| Environment | CQL | ANE | DARC | FOOD (Ours) | | |
| --- | --- | --- | --- | --- | --- | --- |
| | | | | $d_P^\pi$ | $\mu_P^\pi$ | $\nu_P^\pi$ |
| Gravity Pendulum | $0.18 \pm 0.09$ | $-0.22 \pm 0.01$ | $-0.99 \pm 0.55$ | $\mathbf{0.95 \pm 0.03}$ | $-0.17 \pm 0.03$ | $-0.23 \pm 0.01$ |
| Broken Joint Cheetah | $-0.41 \pm 0.03$ | $0.38 \pm 0.03$ | $0.19 \pm 0.1$ | $\mathbf{0.52 \pm 0.05}$ | $0.5 \pm 0.04$ | $\mathbf{0.53 \pm 0.02}$ |
| Broken Joint Minitaur | $-0.53 \pm 0.01$ | $0.03 \pm 0.45$ | $\mathbf{0.41 \pm 0.15}$ | $0.46 \pm 0.29$ | $0.56 \pm 0.22$ | $0.68 \pm 0.34$ |
| Heavy Cheetah | $-0.51 \pm 0.01$ | $\mathbf{0.49 \pm 0.04}$ | $0.02 \pm 0.18$ | $0.14 \pm 0.07$ | $0.15 \pm 0.02$ | $0.13 \pm 0.04$ |
| Broken Joint Ant | $-4.41 \pm 0.17$ | $-2.25 \pm 0.74$ | $-0.47 \pm 0.36$ | $0.03 \pm 0.2$ | $0.6 \pm 0.1$ | $\mathbf{0.65 \pm 0.1}$ |
| Friction Cheetah | $-0.24 \pm 0.01$ | $\mathbf{0.6 \pm 0.18}$ | $0.2 \pm 0.1$ | $0.22 \pm 0.08$ | $0.3 \pm 0.19$ | $0.3 \pm 0.09$ |
| Low Fidelity Minitaur | $0.07 \pm 0.05$ | $-0.13 \pm 0.17$ | $-0.3 \pm 0.1$ | $0.38 \pm 0.15$ | $0.45 \pm 0.1$ | $\mathbf{0.5 \pm 0.05}$ |
| Broken Leg Ant | $-0.24 \pm 0.01$ | $0.16 \pm 0.05$ | $0.01 \pm 0.12$ | $0.1 \pm 0.18$ | $\mathbf{0.17 \pm 0.08}$ | $0.18 \pm 0.15$ |

Table 3: Normalized average return of the RL agents on different Sim-to-Real settings using 5 trajectories from the real environment.

Table 4: Chosen hyper-parameters for both A2C and PPO. The PPO hyper-parameters were fixed for the other environments.

| Hyperparameters | **A2C** | **PPO** |
| --- | --- | --- |
| num-processes | 8 | 8 |
| num-steps | 200 | 1000 |
| lr | $2.5 * 10^{-4}$ | $3.0 * 10^{-4}$ |
| $\gamma$ | 0.99 | 0.99 |
| use-gae | True | True |
| gae-lambda | 0.9 | 0.95 |
| entropy-coef | 0.01 | 0.001 |
| value-loss-coef | 0.4 | 0.5 |
| use-linear-lr-decay | True | True |
| ppo-epoch | N/A | 5 |
| num-mini-batch | N/A | 32 |
| clip-param | N/A | 0.1 |
| TanH Squash | False | True |

**Discriminators training** Both FOOD and DARC incorporate classifiers in their objective. At each epoch, 1000 data points are sampled from both simulated and real transition data sets. The classifiers are then trained with batch sizes of 128 for Pendulum and 256 for the MuJoCo environments. They share the same network structure: a 2 hidden layer MLP with 64 (for Pendulum) or 256 (for MuJoCo) units and ReLU activations. We did not find that the size of the networks played an important role in the results.

### C.4 FOOD HYPER-PARAMETERS SENSITIVITY ANALYSIS

This subsection investigates the impact of our main hyper-parameter $\alpha$, which regulates the strength of regularization that defines a threshold between maximizing the rewards of the simulator and staying close to the real trajectory. All FOOD results are summarized in Figure 3, where, similar to the previous section, FOOD uses the regularization with $d_P^\pi$ in Gravity Pendulum and $\nu_P^\pi$ for the other environments. Note that for the Gravity Pendulum environment, $\alpha \in \{0, 1, 5, 10\}$.

In all the studied environments where PPO was used, we observe that unless for the low or high values of $\alpha$ ($\alpha \in \{0.5, 5\}$), the FOOD agent improves performance compared to $\text{RL}_{\text{Sim}}$. Both cases can be explained. If the value is too high, it may disrupt the gradients and prevent convergence to a good solution. As mentioned in the main paper, this phenomenon also affects the performance in the

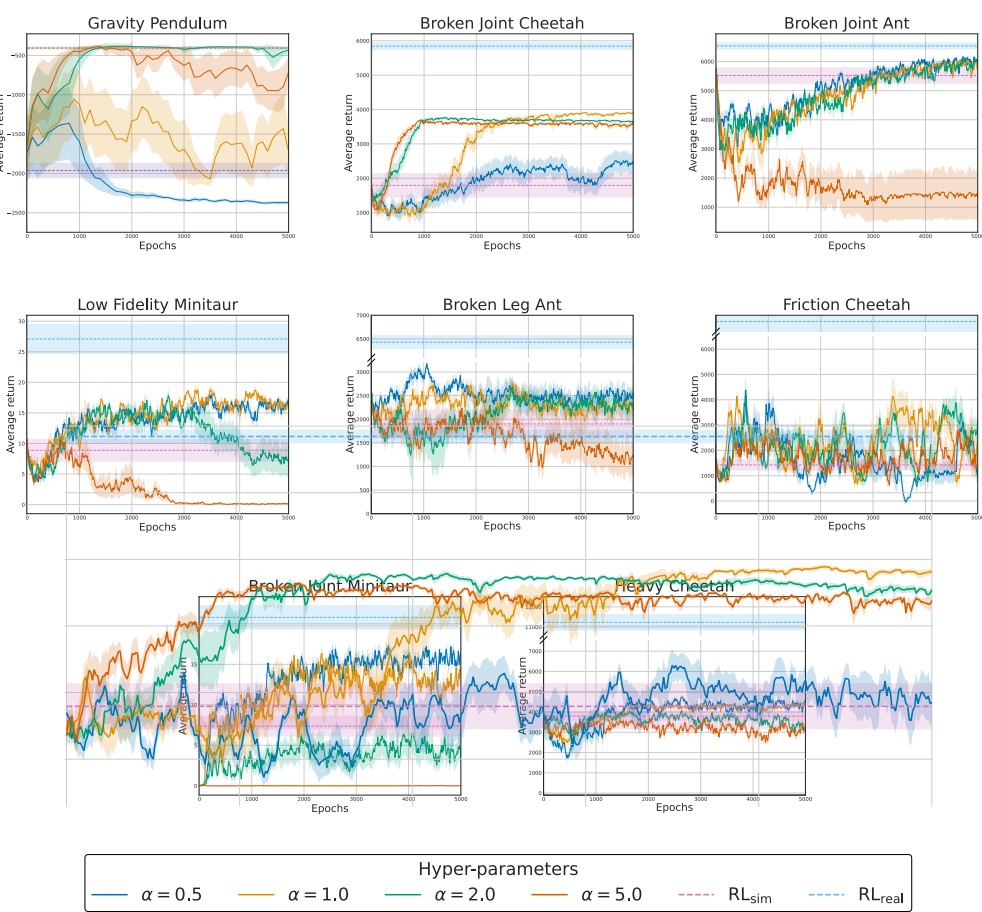

Figure 3: Complete hyperparameter sensitivity analysis for the best FOOD agent on the different Sim-to-Real environments.

simulator, so it would be easy for practitioners to remove such bad hyper-parameters. It may also happen that the strength of the regularization is too low. In that case, FOOD has approximately the same performance as $RL_{Sim}$, as illustrated in Broken Joint HalfCheetah.

Hence, we recommend setting the regularization to have approximately the same weight as the average return. For this, since its advantages are normalized, we recommend using PPO and setting the $\alpha$ parameter to $1$.

## C.5 COMPARISON BETWEEN THE DIFFERENT IL ALGORITHMS FOR THE FOOD AGENT

FOOD is a general algorithm that may use any chosen Imitation Learning algorithm. Each algorithm minimizes a certain type of divergence between state or state-action visitation distributions, as summarized in Table 1. Here, we investigate which IL is better suited for the considered environments.

We compare GAIL (Ho & Ermon, 2016), GAIL-S, GAIL-SAS, AIRL (Fu et al., 2017), PWIL (Dadashi et al., 2020) and PWIL-S in Table 5. GAIL and its extensions were extracted directly from (Kostrikov, 2018), AIRL from (Gangwani, 2021), and PWIL and its extensions were recoded from scratch.

Overall, we observe that all GAIL-associated algorithms have the best results. We attribute this success to the implementation we used, which was optimized for the PPO agent. In addition, FOOD with PWIL has poor results in some environments. This can be attributed to two factors. First, we cannot rule out an error in our code, as we coded it from scratch. Second, this algorithm was

| Environment | GAIL-S | GAIL | GAIL-SAS | AIRL | PWIL-S | PWIL | PWIL-SAS |
|---|---|---|---|---|---|---|---|
| Gravity Pendulum | $-485 \pm 54$ | $-2224 \pm 43$ | $-2327 \pm 14$ | $-1926 \pm 572$ | $-980 \pm 838$ | $-948 \pm 789$ | ?? |
| Broken Joint Cheetah | $3888 \pm 201$ | $3801 \pm 155$ | $3950 \pm 97$ | $3617 \pm 225$ | $3537 \pm 248$ | $2999 \pm 752$ | ?? |
| Heavy Cheetah | $4828 \pm 553$ | $4876 \pm 181$ | $4743 \pm 297$ | $4604 \pm 184$ | $2945 \pm 856$ | $2771 \pm 1235$ | ?? |
| Broken Joint Ant | $5547 \pm 204$ | $6145 \pm 98$ | $6179 \pm 86$ | $5014 \pm 401$ | $3725 \pm 988$ | $3483 \pm 747$ | ?? |
| Friction Cheetah | $3212 \pm 2279$ | $3890 \pm 1495$ | $3852 \pm 733$ | $2957 \pm 1526$ | $3451 \pm 361$ | $3926 \pm 735$ | ?? |
| Broken Joint Minitaur | $13.6 \pm 3.8$ | $14.9 \pm 3$ | $16.6 \pm 4.7$ | $15.8 \pm 2.3$ | $14.6 \pm 1.9$ | $12.1 \pm 5.2$ | ?? |
| Low Fidelity Minitaur | $15.7 \pm 2.8$ | $17 \pm 2$ | $17.9 \pm 0.8$ | $7.5 \pm 5.7$ | $13.6 \pm 5.1$ | $11.4 \pm 3.5$ | ?? |
| Broken Leg Ant | $2345 \pm 806$ | $2652 \pm 356$ | $2733 \pm 719$ | $1634 \pm 857$ | $1490 \pm 714$ | $1554 \pm 886$ | ?? |

Table 5: FOOD sensitivity analysis with respect to the Imitation Learning agent used. We report the average return over $4$ seeds associated with their best hyper-parameter $\alpha$.

introduced in the D4PG agent (Barth-Maron et al., 2018): it is possible that PPO does not leverage well the PWIL's rewards.

An interesting discussion is about GAIL-S, GAIL and GAIL-SAS. The latter focuses on state visitation distributions which should give the FOOD agent more freedom to find a better action. This is for example what is observed in the Gravity Pendulum environment. However, in most cases, we encourage practitioners to use GAIL-SAS, as it gives stable results. This is consistent with both Propositions 4.1 and 4.1: an additional constant appears when only the state visitation distributions are considered.

## C.6 DARC Hyperparameters Sensitivity Analysis

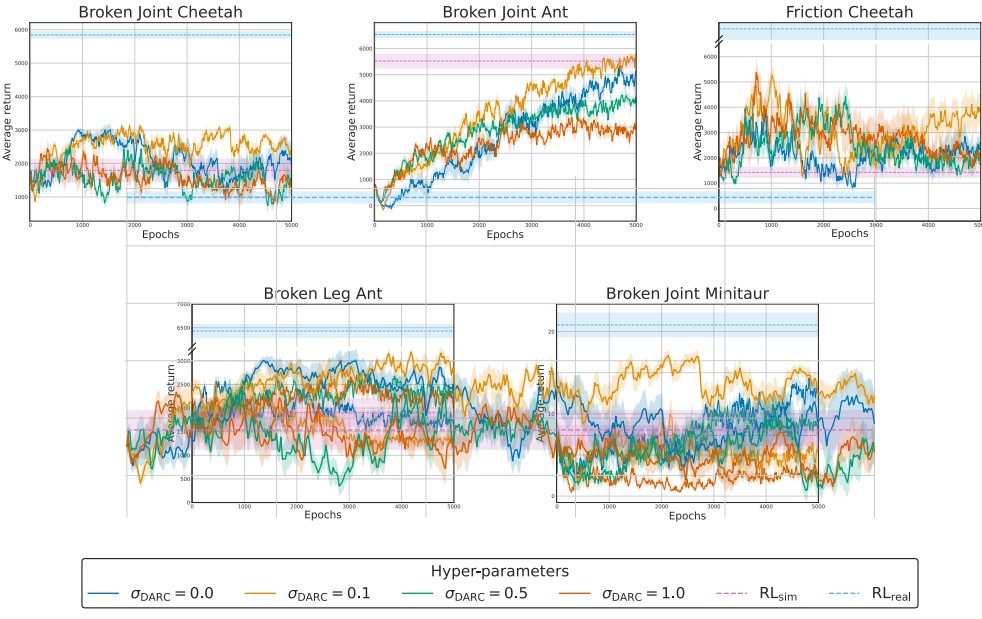

Figure 4: Hyper-parameter sensitivity analysis for the DARC agent on the different Sim-to-Real environments where DARC worked well.

We observe a clear dependence on the noise added to the discriminator, although there seems to be no pattern for choosing the right hyper-parameter. For instance, the best hyper-parameter for Broken Joint Cheetah and Broken Joint Ant is $\sigma_{\text{DARC}} = 0.1$, but this value leads to worse performance than RL$_{\text{Sim}}$ on the two other presented environments.

## C.7  ANE HYPERPARAMETERS SENSITIVITY ANALYSIS

We also detail the ANE's results for all environments. As a reminder, ANE adds a centered Gaussian noise with std $\sigma_{\text{ANE}} \in \{0.1, 0.2, 0.3, 0.5\}$ to the action during training.

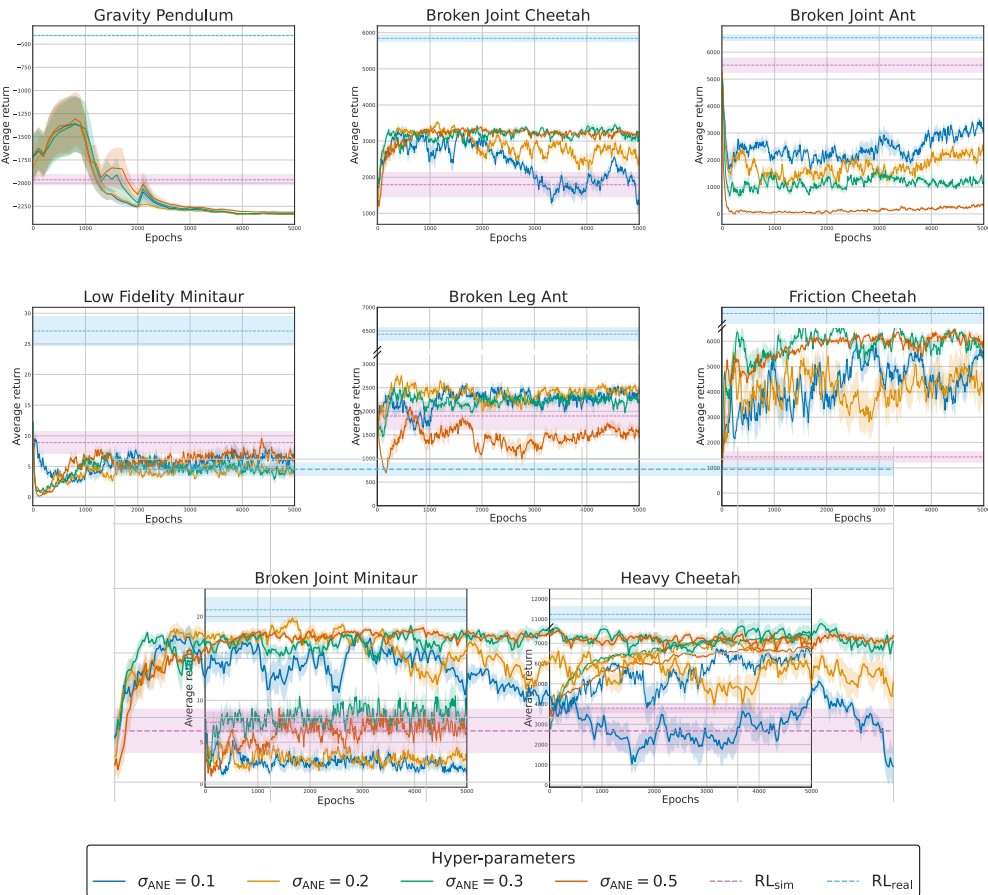

Figure 5: Hyper-parameter sensitivity analysis for the ANE agent on the different Sim-to-Real environments.

These figures are not easily interpretable. This technique may work very well as observed for Heavy Cheetah, but may fail for other environments such as Broken Joint Ant or Low Fidelity Minitaur.