# OpenReview forum: "A Trust Region Approach for Few-Shot Sim-to-Real Reinforcement Learning"
_ICLR.cc/2024/Conference — ICLR 2024 Conference Withdrawn Submission_

### Official Review · Reviewer_mwPX · 2023-10-21

**Soundness:** 2 fair
**Presentation:** 3 good
**Contribution:** 2 fair
**Rating:** 3
**Confidence:** 4

**Summary:**

This paper presents a trust-region-inspired scheme for incorporating real-world data into simulation-based (online) RL. Results and comparisons to relevant baselines illustrate that it can perform well, and a sensitivity analysis provides some insight into the key hyperparameter of the method.

**Strengths:**

- Problem is well-motivated and of great interest to the community
- Literature review is clear and appears to be complete, and in particular points out key ideas which distinguish classes of approaches for this problem and how they relate to the proposed approach
- Experimental construction is clear and well-justified. Experimental analysis is, for the most part, very clear

**Weaknesses:**

- First, a general comment about so-called “trust region” methods in RL: upon reading a standard text on nonlinear programming (e.g., Nocedal and Wright, or Bertsekas), I am increasingly dissatisfied with how the RL community seems to neglect some of the key ideas and defining characteristics of trust region methods. I am not trying to pin blame anywhere, but rather suggest that the authors might do well to try and exploit some of these classical ideas in RL. A few examples:
    - Trust region methods sequentially optimize a model of the true objective built around a nominal point, and at every step, constrain the feasible set to a ball centered at that point. In this paper, the objective in (2) does not appear to be a local approximation to the true objective at current iterate $\theta_k$. Perhaps there is room for improvement here?
    - Likewise, a key idea in the design of good trust region methods is that the size of the feasible set changes at each iteration to reflect how closely the previous “model” matched the true objective. How is $\epsilon_k$ varied across iterations? Does it try to capture this kind of model-mismatch? Perhaps this could be a good direction for improvement.
- nit: this kind of work really reminds me of [1]. While I understand the methods are quite different, the basic problem is the same. I encourage the authors to investigate this literature and discuss connections. (I am not an author of this paper, by the way)
- In 3.1 (first paragraph), the subscript notation on $\mathbb{E}$ could be explained more carefully. This leads to ambiguities later on, e.g., in (1) where my best guess is that s and aanare drawn from distributions d, \pi respectively. But clearly these variables are dependent upon one another, and that dependence is not really very clearly expressed in the notation. One could, for example, imagine that (1) is trying to say that we should sample independent copies of the state and action from their marginal distributions. What is really going on here could be easily clarified by being more precise with notation earlier on.
- Similarly, the role of b_P^\pi is not really made clear when it is first introduced or used in (1), and the reader is left to infer what it is from context and use.
- It seems like a key problem the method will have is the estimation of $d_{P_t}^{\pi_\theta}$ from only a few target domain trajectories. Kernel density estimation is known to be data inefficient, and I am quite skeptical about how the IL methods are going to scale here while remaining accurate. Some discussion here would be good, ideally accompanied by experimental results which highlight performance as a function of the amount of target domain data.
- nit: it is a bit odd that the Nachum 2019 paper is mentioned as coming chronologically after (“then used…”) Touati 2020.
- I do not follow the brief discussion in the sentences above section 4.2.
- Proposition 4.1 does not make much sense to me, although the proof steps are indeed straightforward. What I mean is this:
    - First off, so far as I see, the variable J is never defined anywhere.
    - Relatedly, it is unclear if this result is intended to apply at each iteration of the proposed approach, or to its final result, or even if it has any relationship to the proposed approach (i.e., is it just a general result about sim-to-real?). Relatedly, I suspect (but could not point to a specific paper) that this result is known in the literature since it does not clearly pertain to the proposed algorithm. Perhaps I am just confused. Help me to understand.
    - There is no mention of \epsilon anywhere in the result. Surely the size of the feasible set should influence performance of the proposed algorithm.
- nit: in the caption for table 1, I do not see how one can “minimize the state(-action) visitation distribution” itself. One can minimize a metric of that distribution, but not the distribution itself, right?
- In the experimental protocol paragraph, the bit about removing trajectories that perform poorly in the real environment confuses me. Isn’t that phenomenon exactly what the proposed method is trying to fix? Wouldn’t seeing state transitions that result in poor performance be essential?
    - OR, as I suspect, what is going on here is really that the “trust-region” constraint/regularizer is effectively saying that the simulation has to visit the same states/actions as the real environment (regardless of how much reward they accrued). Something seems strange about this. Comment would be appreciated.
- Dividing the stddev by two in plots is a good indication that results are insignificant… It is also totally statistically useless when computed from 4 random seeds. Why not just show the min and max of all runs? These kinds of statistics are misleading at best -  this kind of thing is common in the literature, and I think it is high time that the community fix such mistakes.
- The discussion below Fig. 1 is confusing to me, and not particularly convincing. For example, how could “a high value induce the agent to replicate real environment trajectories” result in “behavior close to RL_sim?” What am I missing here? More generally, I really don’t see much of a pattern in the figures - it seems like everything is very environment dependent, and as above, I would not draw statistical conclusions from such little data.
- Last, I feel it is a bit unfortunate that a paper on “sim-to-real” did not actually test anything in the real “real world.” Why not try something on a real robot or other setup and put things to the test? I certainly understand that such a test means a lot of work, but it would also go a long way to illustrating the practicality of the proposed method. For instance, there is precious little discussion of the real-world data complexity of the method!

[1] Marco, Alonso, et al. "Virtual vs. real: Trading off simulations and physical experiments in reinforcement learning with Bayesian optimization." 2017 IEEE International Conference on Robotics and Automation (ICRA). IEEE, 2017.

**Questions:**

see above

---

> ### Author Response · Authors · 2023-11-15
> **First answer to Reviewer 4**
>
> Dear reviewer, thank you for your detailed report and your interest in our work. We took your suggestions into account and updated the paper. Here are the answers to your questions.
>
> **General paper update:** We have improved the FOOD agent by considering a regularization between state-action-state visitation distributions. Not only FOOD is now directly aligned with our Proposition, but it also provides better results with lower variance in most environments. Sections 4 and 5 have been updated.
>
> **Robotics application:** We chose to use the term "Sim-to-Real" as we believe "Off-Dynamics RL" is the first step for tackling the real Sim-to-Real problem. We understand that it can be seen as an oversail, but it is quite commonly accepted in the literature [1, 2, 3]. The application of our agent to a real system such as a robot would add an interesting dimension to this work. However, such experiments would be another contribution of its own due to the associated difficulties and costs. We also would like to stress that our experiments are more extensive than the previous works [1, 2, 3], where for instance DARC mainly considers broken environments. For those reasons, we believe the extensiveness of our experiments matches what has been proposed in the Sim-to-Real community.
>
> **Proposition 4.1:** Although simple, to the best of our knowledge, Proposition 4.1. has never been proposed in the literature in the Sim-to-Real setting with a dynamics mismatch, nor in any RL-related literature. We would be happy to reconsider the novelty of our paper if the reviewer can point us to a work that proposes this result.
>
> **Imitation Learning (IL) answers:** IL is known to work with few data samples of the target policy even in high dimensional environments (e.g. Humanoid with $4000$ transitions). Also, our method is agnostic to the IL technique used and will benefit from further advances in IL.
>
> **Experiments:**
> - *Seeds:* Since the experiments are long, having $4$ or $5$ seeds is the norm for RL papers.
> - *Statistical significance:* On the contrary, our algorithm FOOD is the only statistically efficient method compared to the baseline as shown by Table 1 where the standard deviation was not divided by $2$. We performed an unpaired t-test with an asymptotic significance of 0.1 w.r.t. the best performer for each experiment and included it in the rebuttal. We divided the variance by $2$ in Fig. 1 because of the variance of $RL_{\text{sim}}$ which made the figure more difficult to read.
> - *"High value induce the agent to replicate real trajectories":* There may be a misunderstanding here. The high value refers to our hyper-parameter $\alpha$, which controls the amplitude of the regularization. Since PPO's  advantages are normalized, setting $\alpha$ to $5$ forces the agent to focus solely on imitating the transitions from the real environment. It may not always be possible, hence the potential drop in performance.  Unlike other baselines (Cf. Appendix C.5 and C.6), setting $\alpha$ to a value close to $1$ provides a stable algorithm with the strongest results.
>
> **Suboptimal real trajectories:** Due to the particular structure of our method, the quality of the collected trajectories is indeed an important matter. For example, in Broken Joint Cheetah, when collecting $5$ trajectories, the Cheetah falls directly to the ground in one of them. In such cases, practitioners should remove this trajectory as it leads the agent to the wrong parts of the simulator. Nevertheless, in our experiments, we have found that considering medium-quality trajectories for the regularization is sufficient for FOOD to improve over the baselines as it also maximizes the rewards around these trajectories.
>
> **Trust Region term:** We used this term to have a clear continuity with existing algorithms. We understand that it was ill-applied and intend to change it to the more general term "Conservative".
>
> **Proposed paper:** We thank the reviewer for providing us with the related paper "Virtual vs. Real". It combines data coming from two different sources, e.g. a simulator and a real environment, and unlike our baselines and our agent, uses Bayesian Optimization to compute the objective function. We will add this work to the related work section.
>
> **Clarity:** We would like to thank the reviewer again for pointing out the parts of the paper that were unclear. We have taken your suggestions into account and updated the paper.
>
> [1] Desai, S., Durugkar, I., Karnan, H., Warnell, G., Hanna, J.,  Stone, P. (2020) - An imitation from observation approach to transfer learning with dynamics mismatch - NeurIPS.
>
> [2] Eysenbach, B., Asawa, S., Chaudhari, S., Levine, S., Salakhutdinov, R. (2021) - Off-dynamics reinforcement learning: Training for transfer with domain classifiers - ICLR.
>
> [3] Yu, W., Liu, C. K., Turk, G. (2018) - Policy transfer with strategy optimization - ICLR.
>
> [4] Ho, J., Ermon, S. (2016) - Generative adversarial imitation learning - NeurIPS.

---

> > ### Comment · Reviewer_mwPX · 2023-11-22
> >
> > Thank you for the response. I have two direct comments in connection to your note, and a more general wish:
> >
> > - Comment 1: Doing real robot experiments is not a contribution on its own, at least not outside of a field robotics venue like JFR. Rather, it is an opportunity to test a specific scientific hypothesis, e.g., about the efficacy of the proposed method. The effort and expense of conducting such a test may make it infeasible, but not irrelevant. If the authors feel that there is no scientific value in such an experiment, that is one thing, but I see no statement to that effect. In its absence, I stand by my comment and if I am at odds with standards in the community, I will be happy to be overruled by an editor.
> > - Comment 2: My point regarding seeds/statistical significance was that statistics with sample sizes of 4 are utterly meaningless. As above, it should not matter if this is what the community does or not: if we are serious about doing science, then we must care about truth. I am not suggesting that you remove the figures or run more experiments. All I am saying is that the claims should not be statistical in nature - these results cannot support serious hypothesis tests.
> >
> > - A wish: I had hoped to see some more serious engagement with my comments about trust region methods, or at the very least some acknowledgement of the ideas rather than a simple change in nomenclature.

---

> > > ### Author Response · Authors · 2023-11-22
> > > **Second answer to Reviewer 4**
> > >
> > > Dear reviewer mwPX,
> > >
> > > We first would like to address your feedback on the following two points:
> > >
> > > - *Real experiment:* our answer on real-world experiments may have been mis-interpreted. We acknowledge the scientific value of running the agents on a real-world robot. However, due to the associated expenses of conducting such a test, we preferred focusing on benchmarking our approach in various simulation-to-reality conditions across different environments. Similar to previous works, we believe it also allows for a relevant evaluation of the mentioned algorithms.
> > >
> > > - *Statistics with sample sizes of 4 are utterly meaningless:* we conducted an unpaired t-test on our results, taking into account the number of seeds, and found our method to be statistically significant. If you believe this test makes unrealistic hypotheses, we are open to exploring alternative statistical tests, and we would welcome recommendations for tests more suited to our setting.
> > >
> > > Regarding Trust-Region methods, we thank you for pointing us in that direction. We will delve into this literature to improve the theoretical foundation of our agent, and potentially the algorithm itself for a future version of our paper.
> > >
> > > Thanks you for your valuable and constructive feedback.

---

> > > > ### Comment · Reviewer_mwPX · 2023-11-22
> > > >
> > > > Thank you for your response.
> > > >
> > > > - Regarding hardware: "we preferred focusing on benchmarking our approach in various simulation-to-reality conditions across different environments." -> Without hardware, what is being tested is really "simulation-to-different-simulation." I don't want to be an annoyance here, but all I am asking is for a clear discussion in the paper about the scientific hypotheses being evaluated, clear rationale for the construction of experiments, and interpretation of why the results of those experiments support (or invalidate) the hypotheses. Hardware may or may not be needed in this story, but I suspect that a claim like "this method is more reliable for sim-to-real" should be tested "in real life."
> > > > - Regarding small-N t-tests, I urge the authors to consult standard references on the topic. My recollection is that there are assumptions of normality in the underlying populations, similar variances, and more that are surely broken here.

---

### Official Review · Reviewer_iBdd · 2023-10-25

**Soundness:** 2 fair
**Presentation:** 2 fair
**Contribution:** 2 fair
**Rating:** 5
**Confidence:** 4

**Summary:**

This paper focuses on the Sim2Real problem in reinforcement learning. Due to the Sim2Real gap, the well-trained policy from the simulator might perform poorly in real-world scenarios. This paper proposes to constrain the state-action visitation distribution in the real world to be close to that distribution in simulator. Therefore, the policy is optimized for higher returns in the simulator while keeping trajectories that are feasible in the real world. The authors explain the method with theoretical justification, related to trust region approaches (TRPO) and imitation learning approaches (GAIL).

The experiments are conducted on locomotion tasks in OpenAI Gym and Minitaur environment. The proposed method mostly outperforms baselines, including SOTA off-dynamics RL algorithm, SOTA offline RL algorithm and action noise envelope algorithm.

**Strengths:**

The method is well-motivated with simple intuition.

The method is solid and supported by theoretical justification.

The experiments are conducted on many classical RL benchmarks with great performance in comparison with baselines.

**Weaknesses:**

Overall, the technical contribution is not strong enough.

The theoretical part mainly follows derivations from previous paper, such as TRPO. So there is no novel contribution to theory in RL.

The experiments are not extensive enough to fully support the proposed method. See my questions below for more details.

**Questions:**

1. As for the baseline DARC, why is it necessary to re-implement it in the same RL algorithm as the proposed method? It will be great to see how the original DARC (not the modified one) performs in the benchmarks used in this paper.

2. To fix the Sim2Real problem in robotics, domain randomization is an important approach and generally helps improve the robustness of policy trained from the simulator. Is it possible to add this baseline? Logically speaking, if the domain randomization is strong enough to cover the dynamics in the real world, this baseline can perform really well in the evaluation.  I noticed this paper assumes the simulator is a black box. Is this the reason that domain randomization should not be used as a baseline? Why do we need to be constrained by this assumption?

3. The Sim2Real problem is a critical problem in robotics. This paper will be much more impressive if the proposed method can be evaluated in the real world with the real robot.

---

> ### Author Response · Authors · 2023-11-15
> **First answer to Reviewer 3**
>
> Dear reviewer, thank you for your detailed report and your interest in our work. Here are the answers to your questions.
>
>
> **General paper update:** We have improved the FOOD agent by considering a regularization between state-action-state visitation distributions. Not only FOOD is now directly aligned with our Proposition, but it also provides better results with lower variance in most environments. Sections 4 and 5 have been updated.
>
>
> **Modification DARC Baseline:** Similar to DARC, our work focuses on investigating the different objectives that lead to a good transfer between a source and target environment, assuming no access except for a few trajectories in the target environment. Both of our works are agnostic to the underlying RL agent used. In their paper, they chose to optimize it with SAC while we did it with PPO. Their choice was arbitrary as explained in their Section 5. We chose to re-implement DARC in PPO to have a fair comparison between the different objectives. Our FOOD approach could very well have been applied with SAC.
>
> **Domain Randomization Baseline:** Domain randomization is indeed important in the Sim-to-Real setting. To the best of our knowledge, such algorithms are usually difficult to apply when the simulator is a black box. We did not assume to have access to the parameters of the simulator to be as general as possible. In this setting, the baseline RARL [1] is known to drastically fail. ANE, which can be considered a Domain Randomization technique since the inclusion of noise in the action perturbs the transition probabilities, performs well. As explained in the Experiments section, both statements are taken from [2].
>
> **Robotics application:** We chose to use the term "Sim-to-Real" because we believe that "Off-Dynamics RL" is the first step in tackling the real Sim-to-Real problem. It can be seen as an oversailing, but it is widely accepted in the literature [2, 3, 4]. We understand that applying our agent to a real system such as a robot would add an interesting dimension to this work. However, such experiments would be a contribution of its own due to the difficulties and costs involved. We would also like to emphasize that our experiments are more extensive than the previous works [2, 3, 4], where for instance DARC mainly considers broken environments. For example, the Low Fidelity Minitaur benchmark is highlighted as it illustrates a typical situation: the simulator is a simplification of the dynamics of the real world. For those reasons, we believe that the comprehensiveness of our experiments is consistent with what has been proposed in the Sim-to-Real community.
>
>
> [1] Pinto, L., Davidson, J., Sukthankar, R., Gupta, A. (2017, July) - Robust adversarial reinforcement learning - ICML.
>
> [2] Desai, S., Durugkar, I., Karnan, H., Warnell, G., Hanna, J.,  Stone, P. (2020) - An imitation from observation approach to transfer learning with dynamics mismatch - NeurIPS.
>
> [3] Eysenbach, B., Asawa, S., Chaudhari, S., Levine, S., Salakhutdinov, R. (2021) - Off-dynamics reinforcement learning: Training for transfer with domain classifiers - ICLR.
>
> [4] Yu, W., Liu, C. K., Turk, G. (2018) - Policy transfer with strategy optimization - ICLR.

---

> ### Comment · Reviewer_iBdd · 2023-11-20
> **Thank authors for the response**
>
> Thank authors for the response. As for Q1, it will be much better if the proposed method can be combined with SAC, and it outperforms the original DARC. Also, sim2real experiments on real robot will be much more impressive to support your claim. Since my concern is not fully resolved, I'd like to keep my score.

---

> > ### Author Response · Authors · 2023-11-22
> > **Second answer to reviewer 3**
> >
> > Dear reviewer iBdd,
> >
> > We fully take your remarks into account to enhance the quality of our paper in a future version. Specifically, we will add the mentioned additional baselines, and potentially try it on a real world system.
> >
> > Thank you for your insights and constructive feedback.

---

### Official Review · Reviewer_u2wg · 2023-10-29

**Soundness:** 2 fair
**Presentation:** 2 fair
**Contribution:** 1 poor
**Rating:** 3
**Confidence:** 3

**Summary:**

The paper introduces a new sim-to-real transfer algorithm named FOOD for maximizing policy performance in simulation as well as minimizing the state visitation discrepancy between simulation and real environment, so as to achieve high performance considering the dynamics shift. Although the experiments in different simulation environments show the improvement over the DARC algorithm and several other baseline, the proposed trust region method is not novel enough for me as a straightforward combination of RL and IL. The theoretical justification is not directly supporting the proposed algorithm, also the writing is very confusing in several paragraphs, which I think needs further clarification.

**Strengths:**

The paper proposes an effective method for improving sim-to-real (not truly real-world) domain adaptation performances over DARC baseline. The language is good but the description of methods is not clear.

**Weaknesses:**

The novelty of the proposed method is not sufficient. It is a straightforward usage of the trust region method for minimizing the policy state visitation divergence from simulation to real environments.

The definition of $V_{imit}^\pi$ in Eq.5 is not provided and this equation is also not justified with proof.

The Alg.1 is confusing for me without additional details. What does it mean by "select best trajectories’’, and what’s the criteria for the "best’’? What are the objectives for updating the value function and imitation value functions? Where does $\mathcal{D}_t$ and $\mathcal{M}_s$ appear in Eq.4? Please clearly indicate these in papers.

I’m also confused by the CQL baseline, which is an offline-RL algorithm. What does it mean by real-world data and simulator in the paper? CQL is just trained on offline data collected by a behavior policy in simulation, with the objective of maximizing its performance for online evaluation in simulation with the same dynamics. Why does it become a baseline method for sim-to-real setting?

**Questions:**

How is proposition 4.1 related to the practical FOOD algorithms, given that proposition 4.1 shows the bound by state-action-state visitation while in FOOD the imitation learning minimizes the state-action (or state) visitation discrepancy?

---

> ### Author Response · Authors · 2023-11-15
> **First answer to Reviewer 2**
>
> Dear reviewer, thank you for your detailed report and your interest in our work. Here are the answers to your questions.
>
>
> **General paper update:** We have improved the FOOD agent by considering a regularization between state-action-state visitation distributions. Not only FOOD is now directly aligned with our Proposition, but it also provides better results with lower variance in most environments. Sections 4 and 5 have been updated.
>
>
> **Simplicity and novelty:** We do not agree that the complexity of a method is representative of its contribution to the field. On the contrary: if a simple but novel algorithm works in practice then it should be preferred to other more complicated ones. Furthermore, our contribution is not strictly related to our method, but also to the study of a relevant objective to optimize in the Off-Dynamics setting. We have empirically shown that current methods are not general enough for this setting and evaluated the relevance of the baselines and our agent in many scenarios.
>
>
>
> **Writing:** Since the proposition is now directly aligned with our algorithm, we have merged Sections 4.1 and 4.2. We have also corrected the mentioned parts that were unclear:
>
> - *Select best trajectories* is now *Remove trajectories resulting in poor behavior*. Due to the particular structure of our method, the quality of the collected trajectories is indeed an important matter. For example, in Broken Joint Cheetah, when collecting $5$ trajectories, the Cheetah falls directly to the ground in one of them. In such cases, practitioners should remove this trajectory as it leads the agent to the wrong parts of the simulator (hence the original sentence "select best trajectories" in Algorithm 1). Nevertheless, in our experiments, we have found that considering medium-quality trajectories for the regularization is sufficient for FOOD to improve over the baselines as it also maximizes the rewards around these trajectories.
>
> - To clarify Algorithm 1: $\mathcal{D}_t$ is used in the Imitation Learning algorithm to build the imitation reward function, and $\mathcal{M}_s$ is the simulator: the agent only samples in the simulator.
>
> - We are currently justifying the definition of $V^{\pi}_{\text{imit}}$ with proof to justify why it minimizes the JS-divergence between state-action-state visitation distributions when using a discriminator. It will appear in the Appendix in the following days.
>
> - We have not mentioned the equation for updating the value function weights, as it depends on the RL algorithm $\mathcal{O}$ considered. Our agent FOOD is quite general: it focuses on constraining the overall objective to allow a better transfer. This objective can be optimized by any online RL agent that builds a Value function into its learning scheme.
>
> **CQL Baseline:** To properly assess the achievable improvement gained using a simulator in addition to the real data, we consider that Offline Reinforcement Learning (ORL) agents are an important baseline for Few-Shot Sim-to-Real. It is seen as a lower bound to beat. Indeed, the agent has access to some real trajectories. If ORL agents were able to leverage this data to learn a near-optimal policy, then were would be no need for new Sim-to-Real agents. We arbitrarily chose CQL as the ORL agent as it is known to perform well.

---

> > ### Comment · Reviewer_u2wg · 2023-11-22
> >
> > I appreciate the clarification and update by the authors in the rebuttal, especially for the usage of state-action-state visitation instead of state-action visitation for analysis and some details in notations.
> >
> > Now that FOOD with state-action-state visitation outperforms state-action visitation and state visitation, any interpretation on why this is true? This can be an essential discovery for the paper and I'm expecting to see it in future version of the paper.
> >
> > >Simplicity and novelty
> >
> > I'm not saying the paper is not good because of simplicity, but it's straightforward thus lack of novelty.
> > >Select best trajectories
> >
> > This is still a bit odd setting, does the algorithm requiring a human to checkout the trajectories all the time? There should be a quantitative criterion for standardizing the algorithm and its evaluation.
> > >$V_{imit}^\pi$
> >
> > Definition of $V_{imit}^\pi$ is now provided, which is good. Eq. (5) is very important for supporting the proposed algorithm, which still needs to be proved.
> >
> > The writing can be further improved, with more careful handling about notations and references of notations to reduce readers perplexity.
> >
> > Given all above, I remain my score but wish the authors to further enhance the paper.

---

> > > ### Author Response · Authors · 2023-11-22
> > > **Second answer to reviewer 2**
> > >
> > > Dear reviewer u2wg,
> > >
> > > We fully take your remarks into account to enhance the quality of our paper in a future version. Specifically, we plan to:
> > > - Add the proof that the imitation value function minimizes the divergence between the state-action-state visitation distributions.
> > > - Add additional baselines.
> > > - Make the paper clearer.
> > >
> > > Thank you for your insights and constructive feedback.

---

### Official Review · Reviewer_agZp · 2023-11-01

**Soundness:** 3 good
**Presentation:** 3 good
**Contribution:** 2 fair
**Rating:** 5
**Confidence:** 3

**Summary:**

The paper tackles the problem of few-shot sim-to-real. To mimic this, the experimental setting is consider the setting where a source and target simulator is available. The proposed approach is a penalized variant of a trust region approach, where the reward is maximized with an additional term to minimize a divergence between state (action) marginal in the source vs target simulator. As a result, the method resembles GAIL where the critic is trained to distinguish source vs target simulator + the original MDP reward,  rather than generator vs expert as in the original formulation. The approach outperforms DARC, ANE, and CQL baselines on a suite of environments.

**Strengths:**

* The approach is novel to my knowledge.
* The approach is relatively simple and makes sense intuitively in my opinion.
* The approach does not make as many assumptions as related works (e.g. DualDICE's assumption as discussed in section 4.1) and thus can be applied in more settings
* The suite of environments used for evaluation look extensive to me.

**Weaknesses:**

* Using state action of the current policy in the target environment as regularization can harm learning, since a suboptimal policy would result in suboptimal state action distribution used for regularization.
* I am confused as to how this approach can work well in the limited data regime. Doesn't the limited quantity of real data limit the ability for GAIL to learn a good discriminator?
* I'm not so convinced by the baselines. First about CQL. From the text, the authors state "it does not leverage the simulator in its learning process". The text also mentions that "FOOD, DARC, and ANE are trained for 5000 epochs in the simulated environment" but does not mention CQL. This leads me to believe that CQL is only trained on the limited real data, but not on simulated data. In that case, I am not clear on what regularization is used. I would think the most obvious way to implement a CQL baseline is to train on simulated data primarily, and use the real data to reguarlize the Q value. This can be done with online Q learning in the simulated enviornment. I apologize if I am misunderstanding here.
* In a similar vein, simple baselines like TD3+BC can be used, where TD3 is done in simulated environment with BC regularization from real data.

**Questions:**

Could the authors address the points above?

---

> ### Author Response · Authors · 2023-11-15
> **First answer to Reviewer 1**
>
> Dear reviewer, thank you for your detailed report and your interest in our work. Here are the answers to your questions.
>
>
> **General paper update:** We have improved the FOOD agent by considering a regularization between state-action-state visitation distributions. Not only FOOD is now directly aligned with our Proposition, but it also provides better results with lower variance in most environments. Sections 4 and 5 have been updated.
>
>
> **Suboptimal real trajectories:** Due to the particular structure of our method, the quality of the collected trajectories is indeed an important matter. For example, in Broken Joint Cheetah, when collecting $5$ trajectories, the Cheetah falls directly to the ground in one of them. In such cases, practitioners should remove this trajectory as it leads the agent to the wrong parts of the simulator (hence the original sentence "select best trajectories" in Algorithm 1). Nevertheless, in our experiments, we have found that considering medium-quality trajectories for the regularization is sufficient for FOOD to improve over the baselines as it also maximizes the rewards around these trajectories.
>
> **Low data regime:** According to the original GAIL paper [1], this technique is efficient in a low data regime in many high-dimensional environments (sometimes using only hundreds of transitions). In addition, our algorithm is not necessarily limited to GAIL to compute the regularization. It could benefit from future advances in the field that would require even less data than GAIL to be efficient.
>
> **Baselines:**
> - We have considered most of the current methods that fit our setting. DARC is the most prominent approach for tackling the Sim-to-Real gap with few data. ANE is an important baseline for domain randomization when the simulator is a black box [2].
> - To properly assess the achievable improvement gained using a simulator in addition to the real data, we consider that Offline Reinforcement Learning (ORL) agents are an important baseline for Few-Shot Sim-to-Real. It is seen as a lower bound to beat. Indeed, the agent has access to some real trajectories. If ORL agents were able to leverage this data to learn a near-optimal policy, then were would be no need for new Sim-to-Real agents. We arbitrarily chose CQL as the ORL agent as it is known to perform well.
> - A new algorithm named H2O [3] attempts to combine data from the real environment and the simulator in a CQL fashion, as you suggest. However, it is quite clear from their experiments that it requires a lot of real data to be efficient (Cf. Figure 6 of Appendix C.1 of their paper where it requires at least $50$ trajectories to be efficient on HalfCheetah.). We have mentioned this in the related work section, although we are currently making it clearer in a new version of the paper. We can also include this baseline if you think it is still relevant.
> - We argue that TD3 + BC would not be a straightforward baseline to mix the data sampled from both the simulated and real data sets. One could imagine adding the following BC reward to each RL agent:
>
> $$
> r_{\textrm{BC}}^{\mathcal{D}_{t}}(s, a, s') =  \lVert s - s_c \rVert + \lVert a - a_c \rVert + \lVert s' - s'_c \rVert
> $$
>
> where $(s_c, a_c, s_c')$ is chosen as the closest transition of $\mathcal{D}_{\textrm{t}}$ to $(s, a, s')$ with respect to a metric $M$. First, it would require an additional computationally expensive step to select this transition. Second, this bonus would be subject to compounding errors: it only considers one-step transitions, not the whole trajectory induced by the real environment. Our agent FOOD, on the other hand, encompasses the whole real trajectories induced by the real dynamics in its regularization and is easy to implement.
>
> [1] Ho, J., Ermon, S. (2016) - Generative adversarial imitation learning - NeurIPS.
>
> [2] Hanna, J. P., Desai, S., Karnan, H., Warnell, G., Stone, P. (2021) - Grounded action transformation for sim-to-real reinforcement learning - Machine Learning.
>
> [3] Niu, H., Qiu, Y., Li, M., Zhou, G., HU, J., Zhan, X. (2022) - When to trust your simulator: Dynamics-aware hybrid offline-and-online reinforcement learning - NeurIPS.

---

> > ### Comment · Reviewer_agZp · 2023-11-22
> > **Thanks for your response.**
> >
> > Thank you for clarifying about low data regime. I guess it can be both a pro and a con that this method relies on on how data efficient GAIL (or other future IL algorithm) is. To me this is one of the more important aspects of the algorithm. On a related note, the authors mention that H20 requires more real data to be efficient. I believe a more thorough empirical investigation of how the method and baselines (including H20) scale with the amount of real data. I think that would make a more convincing case for FOOD.
> >
> > As for TD3+BC, I was imagining something far simpler, which is just to apply online TD3 in simulation, and BC loss on the real state transitions. Regardless, I don't think this point is as important, and was more of a minor suggestion for additional simple baselines to make FOOD more convincing.
> >
> > Finally, I'd like to comment on some of the discussion I see raised in the other threads.
> > * Real robot experiments: While I do not think its necessarily a pre-requisite, I agree with the other reviewers that this would strengthen the paper. While existing sim-to-real papers have been published without real robot experiments, I believe that as the field progresses the bar should continue to be raised, and real robot experiments may become more important in ensuring that we are not overfitting to toy environments.
> > * Statistical significance: I'm not sure about reviewer mwPX's claim that "statistics with sample sizes of 4 are utterly meaningless" when the authors seem to have performed an unpaired t-test in the updated paper. I am interested in hearing the reviewer's thoughts on why this test does not convince them.
> > * I found the comments regarding trust region approaches by Reviewer mwPX as very insightful and also wish the authors had engaged more deeply here.
> >
> > With all this taken into context, I will keep my score. I will improve my score if  the following are met:
> > * More extensive experimental evaluation (including original SAC-based DARC and comparison with H20, as a function of real data size, etc.). At this stage a real robot experiment is obviously too late but I would encourage the authors to explore this for the future.
> > * More theoretical engagement/incorporation of the ideas proposed by Reviewer mwPX

---

> ### Author Response · Authors · 2023-11-22
> **Second answer to Reviewer 1**
>
> Dear reviewer agZp,
>
> We fully take your remarks into account to enhance the quality of our paper in a future version. Specifically, we plan to:
> - Add the additional baselines H2O, a modification of TD3+BC for our setting, and try SAC-based DARC on our environments.
> - Add additional theory from Trust Region methods to back up the relevance of our algorithm and potentially improve it.
> - Potentially add a robot experiment if it is possible.
>
> Thank you for your insights and constructive feedback.

---

### Meta-Review · Area_Chair_UavQ · 2023-12-05

**Metareview:**

Summary: The paper proposes a trust-region-inspired method, named FOOD, to address the Sim2Real gap in reinforcement learning. It aims to optimize policy performance in simulation while minimizing the state-action visitation distribution difference between simulation and the real world. The proposed method demonstrates improvements over baseline algorithms in locomotion tasks across various environments.

Weaknesses: Reviewers express concerns about its novelty, clarity in presentation, and the extent of experimental validation. Some reviewers question the choice of baselines, particularly CQL, and suggest additional clarity in defining and justifying key components of the method. Reviewer mwPX made a key point about labeling this a trust region method, which was not satisfactorily addressed by authors. Overall, the paper receives mixed feedback on its contribution, presentation, and experimental validation.

Strengths: The problem setting is interesting and important, the approach is somewhat well-motivated, and the experimental design clear and well-constructed.

Suggestions: Addressing the several raised concerns and providing additional explanations could enhance the paper's clarity and impact.

**Justification For Why Not Higher Score:**

Reviewers express concerns about its novelty, clarity in presentation, and the extent of experimental validation. Some reviewers question the choice of baselines, particularly CQL, and suggest additional clarity in defining and justifying key components of the method. Reviewer mwPX made a key point about labeling this a trust region method, which was not satisfactorily addressed by authors. Overall, the paper receives mixed feedback on its contribution, presentation, and experimental validation.

**Justification For Why Not Lower Score:**

N/A

---

### Decision · Program_Chairs · 2024-01-16

Reject